# Harnessing indole scaffolds to identify small-molecule IRE1α inhibitors modulating *XBP1* mRNA splicing

Yang Liu [1,2,3,5], Amrutha K. Avathan Veettil [1,2,3,5], Raphael Gasper [4], Mao Jiang[1,2,3], Leon Wagner[1,2,3], Oguz Hastürk[1,2,3] & Peng Wu [1,2,3] ✉

The inositol-requiring enzyme 1 alpha (IRE1α) is an important sensor protein with dual kinase and ribonuclease function. It induces *X-box binding protein 1* (*XBP1*) mRNA splicing and mediates endoplasmic reticulum (ER) stress-triggered downstream unfolded protein response signaling pathways. The dysregulation of IRE1α has been associated with multiple human diseases, and thus IRE1α-targeting small molecules harbor great therapeutic potential. We herein report a series of substituted indoles as IRE1α inhibitors (such as IA107) of excellent potency and selectivity. We also report a resolved co-crystal structure that reveals a unique inhibition mode of IA107 that allosterically inhibits IRE1α RNase activity via binding to the IRE1α kinase domain but without inhibiting the IRE1α dimerization. The following cellular evaluation results demonstrate that IA107 concentration-dependently inhibits the cellular ER stress-induced *XBP1* mRNA splicing, and the ester-containing prodrug exhibits a ~ 50-fold increase in cellular activity. Collectively, our results establish the indoles as a potent and selective IRE1α-inhibiting chemotype that modulates RNA splicing and expands the biological application potential associated with IRE1α targeting via small molecules.

Accumulation of misfolded protein in the endoplasmic reticulum (ER) lumen triggers stress, which activates the conserved unfolded protein response (UPR). UPR mediates ER function through three distinct ER transmembrane sensors including inositol-requiring enzyme 1 (IRE1)[1–4]. Dysregulation of the UPR signaling is associated with the genesis and progression of many human diseases including cancers[5–8], neurodegenerative diseases[9,10], metabolic disorders[11,12], and pain[13,14]. Thus, regulating the UPR signaling activity by targeting the transmembrane sensors poses a promising strategy to address human diseases. The human genome encodes two IRE1 isoforms, IRE1α and IRE1β[15]. IRE1α, the most evolutionarily conserved UPR sensor that is ubiquitously expressed, consists of the N-terminal ER luminal domain that senses unfolded proteins[16,17] and the C-terminal

cytoplasmic region that includes the serine/threonine kinase domain and the endoribonuclease (RNase) domain, exhibiting dual kinase and endoribonuclease functions[18]. IRE1α oligomerization and autophosphorylation under ER stress activate its endoribonuclease activity via binding and cleaving RNA substrates, triggering downstream UPR pathways[19,20]. For example, IRE1α induces splicing of the *X-box-binding protein 1* (*XBP1*) mRNA that encodes the transcription factor XBP1 spliced isoform protein (XBP1s)[21,22], which upregulates genes involved in the ER protein folding and secretion, promotes degradation of misfolded proteins, and further relieves ER stress[18]. In addition to the IRE1–*XBP1* pathway, the regulated IRE1-dependent decay (RIDD) serves as an avenue to mediate ER stress, in which IRE1α-mediated RNA cleavage leads to decreased

[1]Chemical Genomics Centre, Max Planck Institute of Molecular Physiology, Dortmund, Germany. [2]Department of Chemical Biology, Max Planck Institute of Molecular Physiology, Dortmund, Germany. [3]Faculty of Chemistry and Chemical Biology, TU Dortmund University, Dortmund, Germany. [4]Crystallography and Biophysics Unit, Max Planck Institute of Molecular Physiology, Dortmund, Germany. [5]These authors contributed equally: Yang Liu, Amrutha K. Avathan Veettil. ✉e-mail: peng.wu@mpi-dortmund.mpg.de

mRNA abundance and lowered the load of protein folding (Fig. 1a)[23]. IRE1α also interacts with other protein pathways to modulate various cellular processes, including cell death and autophagy[24–26]. Given its role of being a key effector in the UPR pathway in maintaining ER homeostasis and cell fate, as well as its involvement in the progression of various diseases, IRE1α is a promising target for therapeutic intervention in cancers, immune diseases, and metabolic and neurodegenerative disorders[3,27]. Subsequently, the development of IRE1α-targeting small molecules has attracted gaining interest in recent years.

IRE1α-inhibiting small molecules can be broadly classified into kinase domain binders (target ATP-binding pocket of the kinase domain) and RNase domain binders (target the catalytic core of the RNase domain), with selected examples shown in Fig. 1b[28–37]. Following the classification for small-molecule kinase inhibitors, the kinase

domain binders can be mainly classified as type I and type II inhibitors[28,38], including the imidazo[1,5-a]pyrazin-8-amine compounds 3, KIRA7 and KIRA8 (AMG-18)[28–30,39,40]. In addition to the type I and II inhibitors, GSK2850163 was reported as a type III IRE1α inhibitor that occupied a pocket next to the hinge region of the kinase[32,40], and the imidazo[1,2-b]pyridazin-8-amine compound 31 was reported to bind IRE1α via an allosteric mechanism involving an unusual "DFG up" conformation with the substantial disordering of the αC-helix[41]. As for the RNase domain binders, a series of inhibitors sharing the hydroxy-aryl-aldehyde scaffold were reported to bind directly to the catalytic core of the IRE1α RNase domain[34–37]. Despite the reported IRE1α inhibitors with impressive potency and encouraging progress, unexplored inhibitor chemotypes with distinct inhibitory mechanisms may offer alternative therapeutic options in addressing related diseases. Consequently, there is a considerable unmet need to develop IRE1α-targeting

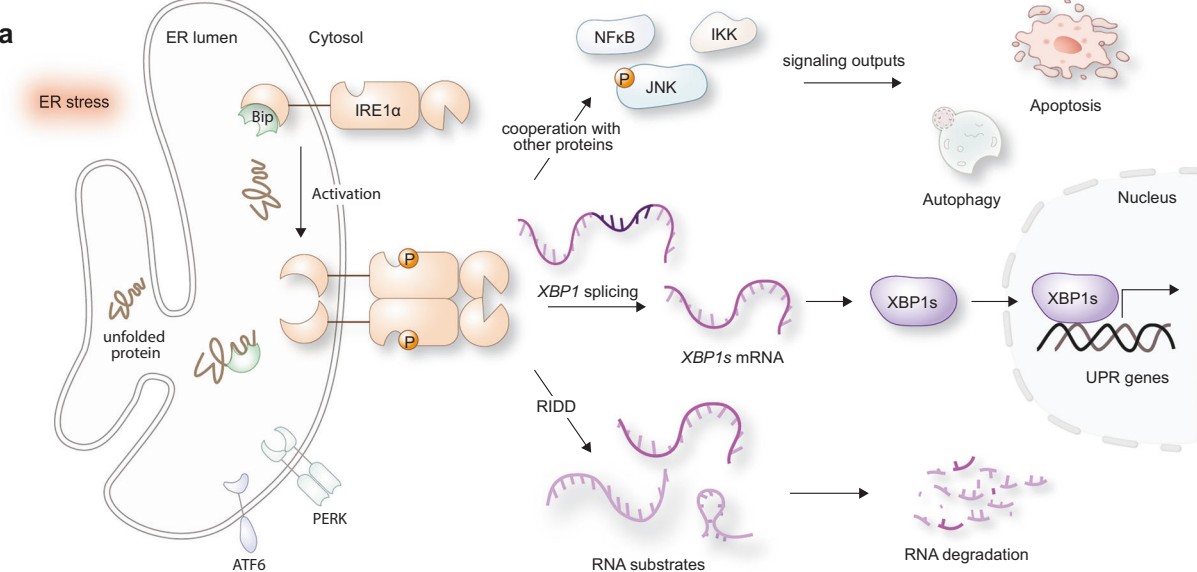

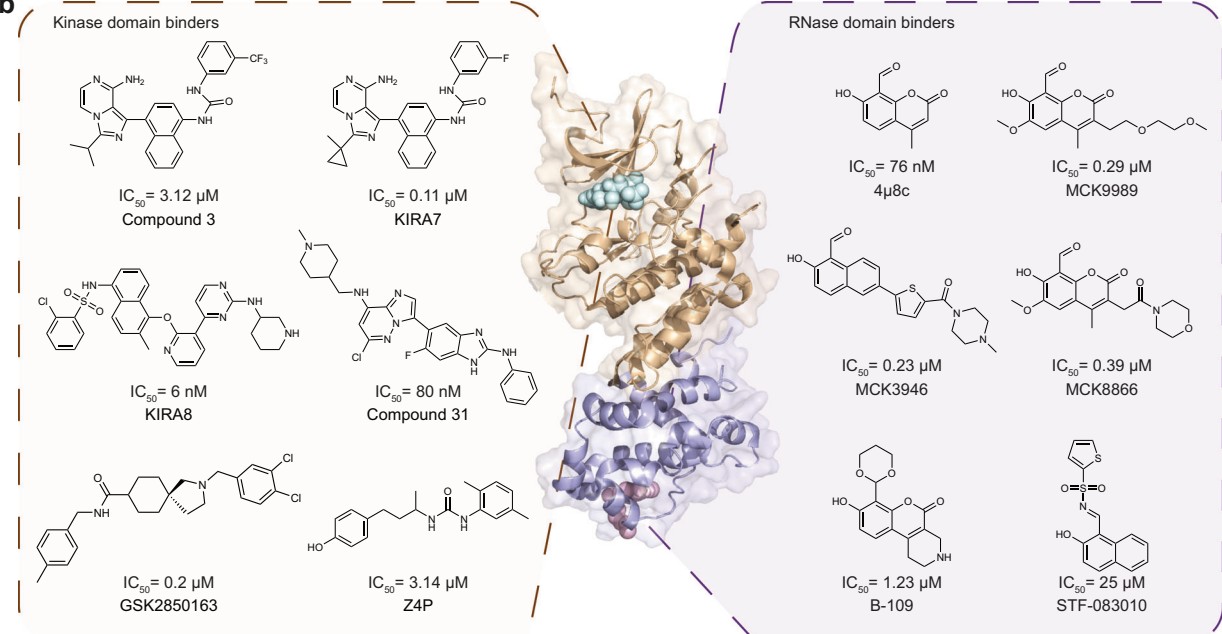

**Fig. 1 | IRE1α pathway and reported IRE1α RNase inhibitors. a** Illustration of the IRE1α pathway showing the three main downstream effects, the *XBP1* splicing, the induction of apoptosis and autophagy, and the regulated IRE1α-dependent decay (RIDD). ER endoplasmic reticulum. UPR unfolded protein response. **b** Examples of reported IRE1α inhibitors targeting either the kinase domain or the ribonuclease domain (PDB ID: 4PL3).

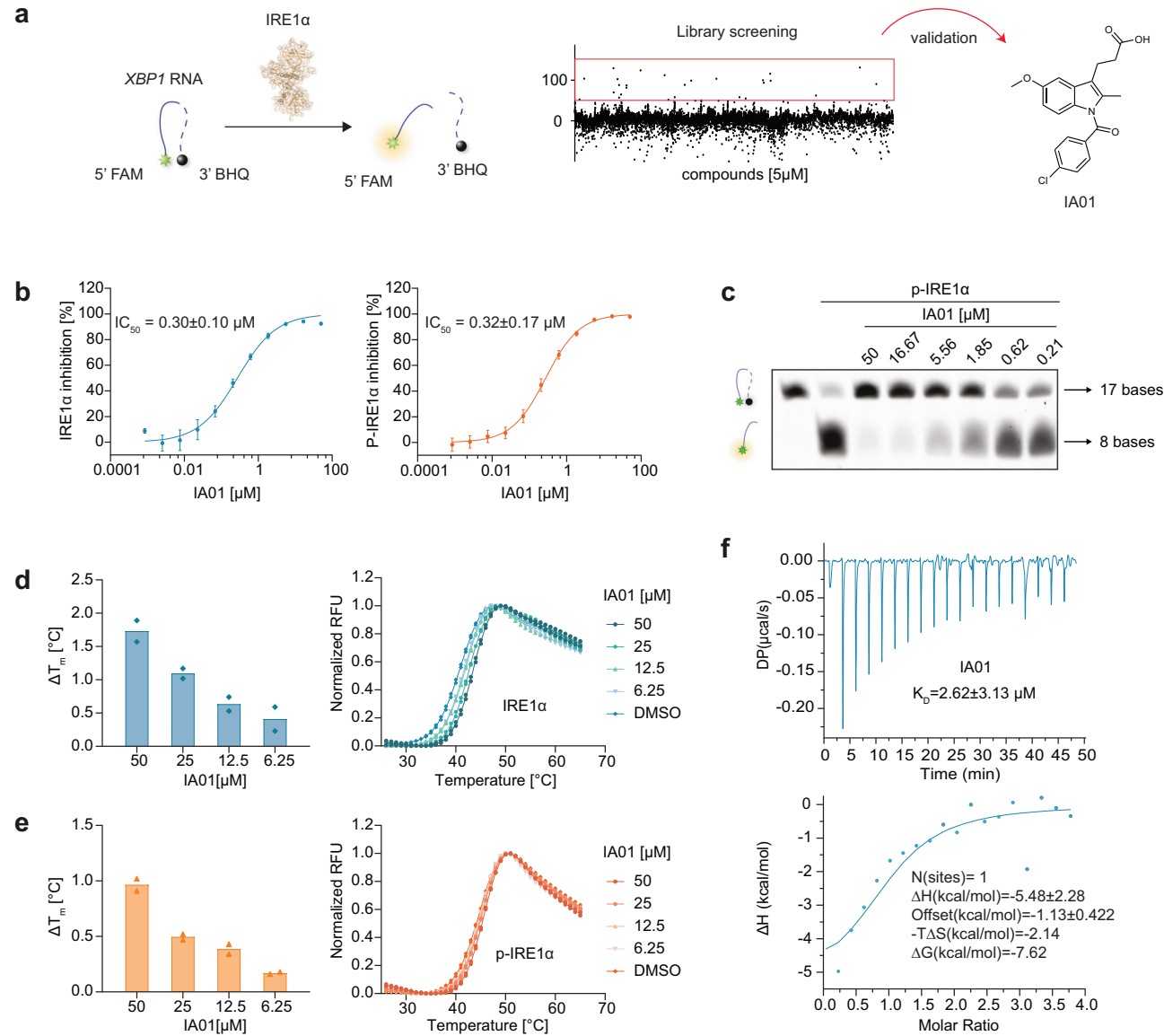

**Fig. 2 | Discovery of the indole-based IRE1α RNase inhibitor IA01. a** The identification of IA01 as an IRE1α inhibitor hit using a FRET assay. **b** IA01 inhibited IRE1α and phosphorylated IRE1α RNase activity in the FRET-based assay, with IC$_{50}$ of 0.30 ± 0.10 μM and 0.32 ± 0.17 μM, respectively. Data are presented as mean ± SEM. $n$ = 3 for IRE1 measurement, $n$ = 4 for p-IRE1α measurement. **c** IA01 concentration-dependently inhibited phosphorylated IRE1α RNase activity in a gel-based cleavage assay. The expected number of bases was labeled. Representative result of $n$ = 3. **d** IA01 stabilized IRE1α concentration-dependently in the DSF assay. $n$ = 2. **e** IA107 stabilized phosphorylated IREα concentration-dependently in the DSF assay. Left panels, $\Delta T_m$; right panels, melting curve. $n$ = 2. **f** IA01 bound to IRE1α with a $K_D$ of 12.0 ± 9.81 μM (by titrating 200 μM IA01 into 10 μM IRE1α) measured by ITC assay. IRE1α dephosphorylated IRE1α, p-IRE1α phosphorylated IRE1α.

inhibitors with diverse chemical scaffolds and distinct mechanisms of inhibition.

In this study, we report a class of highly potent and selective IRE1α inhibitors based on an indole scaffold. We performed an in-house screening of a small-molecule library with scaffold-diverse compounds and discovered indole-based compounds as IRE1α inhibitors. The subsequent pipeline of systematic structure modifications, extensive biochemical and biophysical characterizations, and the successful resolution of a co-crystal complex structure, together with potent cellular performance against mRNA splicing using a prodrug strategy, led to the identification of the indole-based IRE1α inhibitors with potent activity and high selectivity. Broadly, in addition to providing potential therapeutic chemotypes, inhibition of the IRE1–UPR pathway via the validated indole small molecules will assist in probing the diverse downstream effects associated with the UPR signaling network.

## Results and discussion

### Identification and characterization of the IRE1α hit IA01

To initiate the discovery pipeline for the identification of IRE1α inhibitors, a fluorescence resonance energy transfer (FRET)-based assay was established as the primary screening assay by using the dual FAM- and BHQ-labeled *XBP1* RNA hairpin and the truncated IRE1α protein covering the dephosphorylated kinase domain and the RNase domain (547–977, refer as IRE1α in this study, Supplementary Fig. 1a). Upon screening an in-house library of ~10,000 compounds at a single concentration of 5 μM (Supplementary Fig. 1b), the indole-containing IA01 was identified as the most promising hit among the screening set (Fig. 2a). The phosphorylation of IRE1α plays a crucial role in exerting its ribonuclease activity and given that some compounds inhibited the phosphorylated IRE1α while in parallel activated the dephosphorylated IRE1α,[42] in this study, both phosphorylated and dephosphorylated IRE1α were evaluated for the inhibitory activity with the indole

inhibitors (Supplementary Fig. 1c, d). IA01 showed equivalent inhibitory activity towards dephosphorylated IRE1α and phosphorylated IRE1α (p-IRE1α) with IC$_{50}$ being 0.30 μM and 0.32 μM, respectively (Fig. 2b). The inhibitory activity was then validated in a gel-based cleavage assay using phosphorylated IRE1α (Fig. 2c). In the differential scanning fluorimetry (DSF) assay, IA01 concentration-dependently stabilized dephosphorylated IRE1α (Fig. 2d) and phosphorylated IRE1α (Fig. 2e), indicating the direct binding between IA01 and IRE1α. The binding affinity between IRE1α and compound IA01 was further confirmed in the isothermal titration calorimetry (ITC) assay with a $K_D$ of 12 μM (Fig. 2c).

### Structural modifications yielding potent IRE1α inhibitor IA107

Given the structural similarities among IA01, indomethacin, and auxin, we first evaluated indomethacin and auxin analogues available from our in-house small-molecule collection. The results indicated that neither indomethacin and auxin nor their analogs showed inhibition activities of more than 50% at the tested concentration of 50 μM (Supplementary Fig. 2a). Subsequently, to thoroughly examine the structure-activity relationships surrounding the indole scaffold of IA01, we performed structural modifications at the different substituents ($R^1$, $R^2$, $R^3$) to probe pharmacological features that would be required to improve IRE1α inhibitory activity.

First, the propionic acid group at the C-3/$R^1$ position was replaced by other carboxylic groups of different chain lengths, alkyl groups, esters, a ketone, aromatic rings, as well as an aliphatic heterocyclic ring[43]. As illustrated in Fig. 3a, the most potent activities were observed when the $R^1$ groups were carboxylic acid derivatives. In comparison, all other modifications resulted in a great loss of activity (IRE1α inhibition <30%), indicating that structural modification at the indole C-3 position was not tolerated beyond the carboxylic acid groups. Hence, we retained the propionic acid at the C-3/$R^1$ position (IA01 IRE1α, IC$_{50}$ 0.30 ± 0.10 μM) in the following studies.

Next, modification at the $R^2$/N-substituent position showed that a variety of acyl groups can be accommodated (Fig. 3b), including substituted phenyl rings, aromatic heterocyclic rings, aliphatic rings, as well as fused or isolated bicyclic and tricyclic rings. The N-benzoyl group with different electron-withdrawing and -donating groups led to only subtle variations in the inhibitory activity. In contrast, the unsubstituted N-benzoyl analogue (IA64) exhibited a significant activity improvement (IRE1α, IC$_{50}$ 0.030 ± 0.01 μM). Encouraged by the result, we proceeded to obtain the indole analogues substituted by thiophene, benzothiophene, and nitrogen-containing heterocycles, most of which retained the IRE1α inhibitory activity compared to that of the phenyl-substituted IA64. In contrast, the replacement of the aromatic rings with a secondary amino moiety (IA76) or complete removal of the acyl group (IA72) led to a complete loss of activity (IRE1α, IC$_{50}$ > 20). Lastly, given the potent inhibitory activities of the thiophene-2-carbonyl IA03 and the 3-cyanobenzoyl IA30, we made a mini-collection of the corresponding analogues by replacing the 5-methoxy with different halogenated groups, leading to compounds with mostly retained IRE1α inhibitory potency (Supplementary Fig. 2b, c).

Third, based on the structural features and activities of IA01 and IA64, we continued with structural modification at the $R^3$/C-5 position by retaining the $R^1$ as the carboxylic acid and the $R^2$ as the benzoyl group (Fig. 3C). Indoles with a phenyl group substituted by electron-withdrawing or -donating groups (IA 149 and IA 146) at the *meta*-position displayed equivalent potency (IRE1α IC$_{50}$ 1.01 ± 0.27 μM and 1.21 ± 0.82 μM), while *ortho*- and *para*-substituted aromatic ring led to dramatically decreased potency. Replacing the methoxy group of IA64 with a trifluoromethoxy group (IA148, IRE1α IC$_{50}$ 0.57 ± 0.36 μM) led to a ~19-fold reduction in IRE1α inhibition. Unsubstituted indole C-5 retained the activity that was 10-fold less than that of IA64. A remarkable improvement in potency was observed when halogen

substitutions were introduced at the $R^3$ position, e.g., IA107 with a bromide at the C-5 position showed a 20-fold increase in potency (IRE1α, IC$_{50}$ 0.016 ± 0.001 μM), the most potent inhibitor among the synthesized collection.

### Biochemical and biophysical characterization of IA107 and the binding mechanism studies

IA107 was tested as the most potent inhibitor among the structurally modified analogues with IC$_{50}$ being 16 nM and 9 nM against IRE1α and p-IRE1α, respectively (Fig. 4a). To validate the inhibitory activity in orthogonal assays, we proceeded with the gel-based RNA cleavage assay, the results of which indicated that IA64 (Supplementary Fig. 3a) and IA107 (Fig. 4b) concentration-dependently inhibited p-IRE1α RNase activity. The binding of the indole inhibitors to IRE1α was confirmed via the differential scanning fluorimetry (DSF) assay using a naphthalene-containing compound IA34 as a negative control (IRE1α and p-IRE1α IC$_{50}$ > 20 μM) (Supplementary Fig. 3b). The DSF results showed that IA03, IA30, IA64, IA107, IA141, and IA147 concentration-dependently stabilized IRE1α upon binding. The observed changes in the melting temperature ($\Delta T_m$) correlated with the inhibitory activities tested in the biochemical assay. The negative control compound IA34 did not show any detectable changes (Fig. 4c and Supplementary Fig. 3c–e). In addition to IRE1α, the compounds were evaluated against p-IRE1α, for which consistent results were observed in the DSF (Fig. 4d and Supplementary Fig. 3f–h). Compounds IA64, IA107, IA141, IA147 bound to IRE1α with $K_D$ of 2.96 μM, 0.94 μM, 1.82 μM, 2.99 μM, respectively, obtained in ITC (Fig. 4e and Supplementary Fig. 3i, j). Furthermore, the binding affinity of compound IA107 was confirmed by the microscale thermophoresis (MST) assay, which gave a $K_D$ of 1.36 μM (Fig. 4f). The binding affinity of compound IA107 against p-IRE1α was measured by MST, which showed an increased affinity with a $K_D$ of 152 nM. The different binding affinity ($K_D$) against different IRE1α states could be explained by that IA107 prefers to bind to the dimeric form of IRE1α to inhibit its RNase activity. The p-IRE1α assembles more dimers and therefore has a higher affinity to IA107 in comparison with unphosphorylated IRE1α. This may also explain the observed discrepancy between the inhibitory IC$_{50}$ and the binding $K_D$ of IA107. Once the dimeric proteins are bound and inhibited, the RNase activity of IRE1α will be inhibited since the monomeric proteins are inactive[44,45].

To investigate the inhibition mechanism of the indole inhibitor, the Michaelis–Menten kinetic analysis of IA107 was performed to determine whether IA107 competes with the *XBP1* RNA substrate (Fig. 5a), which showed that only minor changes of the $K_m$ were observed under different concentrations of IA107. As the $K_m$ for the DMSO control was 876.5 nM, and the $K_m$ for IRE1α incubated with 0.01 μM, 0.1 μM, and 1 μM IA107 were 915.5 nM, 819.2 nM, and 1210 nM, respectively. In contrast, the calculated $V_{max}$ decreased in a concentration-dependent manner, as the $V_{max}$ for the DMSO control, 0.01 μM, 0.1 μM, and 1 μM IA107 were 10.98 FL. I/s (fluorescence intensity/second), 7.98 FL. I/s, 2.33 FL. I/s and 0.9 FL. I/s, respectively. Thus, the overall constant $K_m$ and concentration-dependent $V_{max}$ suggested a non-competitive inhibition mechanism towards the *XBP1* mRNA substrate.

Given that small molecule binding to the ATP-binding site of the IRE1α kinase domain could allosterically inhibit or activate the RNase activity of IRE1α[28,32], we continued with the investigation of the potential binding mode of IA107 at the ATP-binding site of the IRE1α kinase domain. Here, a previously reported IRE1α RNase activator G1749 binding to the ATP-binding site of IRE1α kinase domain[42] was used as a reference compound (Supplementary Fig. 4a). We first proceeded with the ITC measurement, which showed that the binding affinity of G1749 decreased concentration-dependently in the presence of IA107, e.g. G1749 did not show a detectable binding with IRE1α in the presence of 1 mM IA107 (Fig. 5b). In comparison, the competition

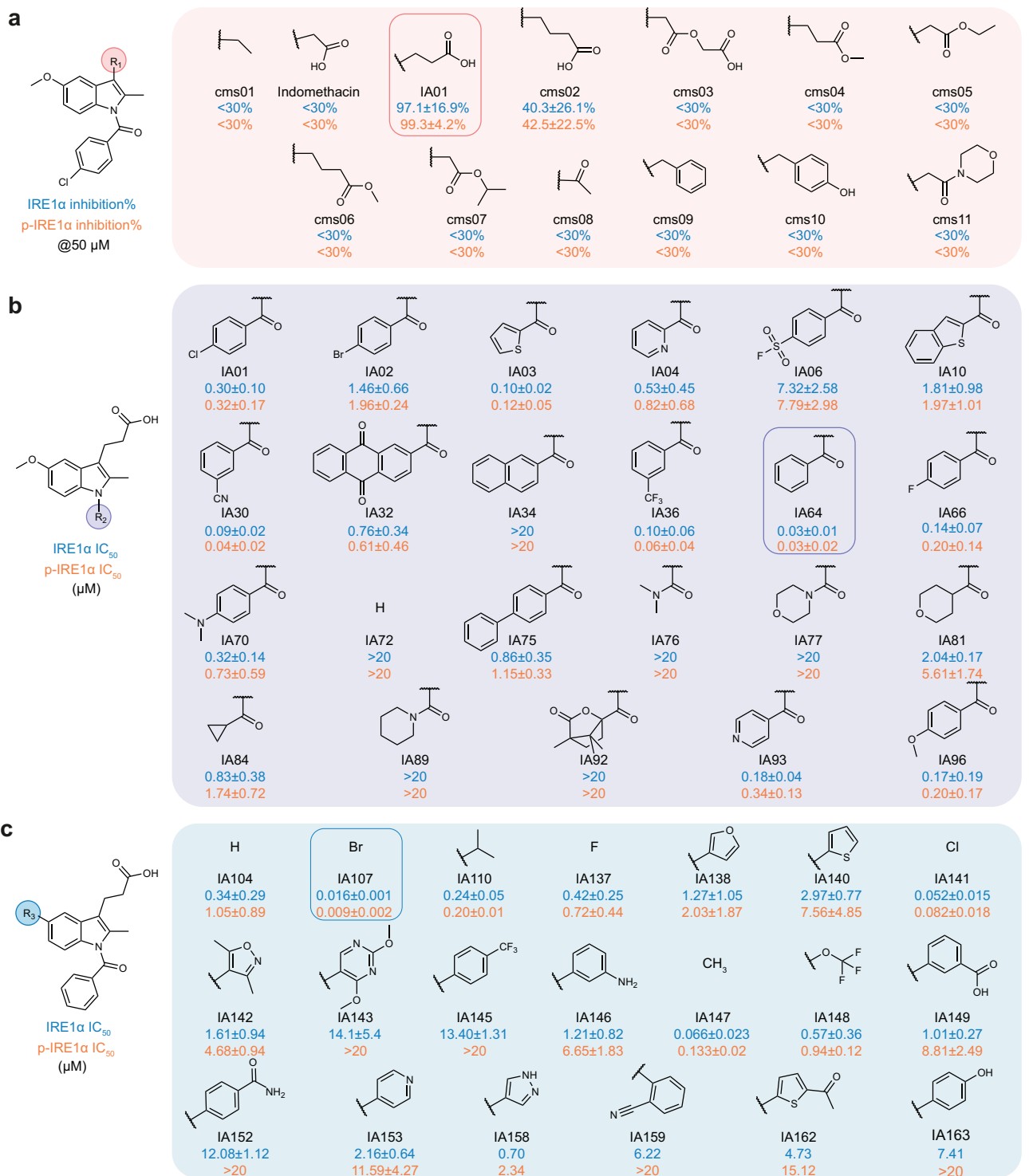

**Fig. 3 | Overview of the evaluated indole analogues as IRE1α inhibitors. a** Indole analogues with different substituents at the C-3position (R¹) and inhibition (%) activity tested in FRET assay at 50 μM. Data are presented as mean ± SEM, *n* = 3. **b** Indole analogues with different substituents at the N-position (R²). Data are presented as mean ± SEM. *n* = 3 for most of the compounds except IA10, IA32, IA64 against IRE1α and IA01, IA 64 against p-IRE1α measurements, *n* = 4. **c** Indole analogues with different substituents at the C-5 position (R³). The inhibitory data against the unphosphorylated IRE1α tested in the FRET assay are shown in blue fonts, and the inhibitory data against the phosphorylated IRE1α tested in the FRET assay are shown in orange fonts. Data are presented as mean ± SEM. *n* = 3 for all measurements except compounds IA158, IA159, IA162, IA163, *n* = 2.

ITC measurement by pre-incubating IRE1α with the negative control compound IA34 (IA34 showed no binding signal with IRE1α in ITC, Supplementary Fig. 4b) showed no significant impact on G1749 binding to IRE1α (Fig. 5c and Supplementary Fig. 4c). IA107 affected the binding of G1749 with IRE1α and the binding of G1749 can compete with that of IA107, indicating that IA107 and G1749 share the same binding site at the IRE1α kinase domain. The binding of IA107 to the ATP-binding site of the kinase domain was further confirmed by titrating IA107 to IRE1α pre-incubated with different concentrations of G1749 (Fig.5d, Supplementary Fig. 4d). IA107 showed a $K_D$ of 0.94 μM

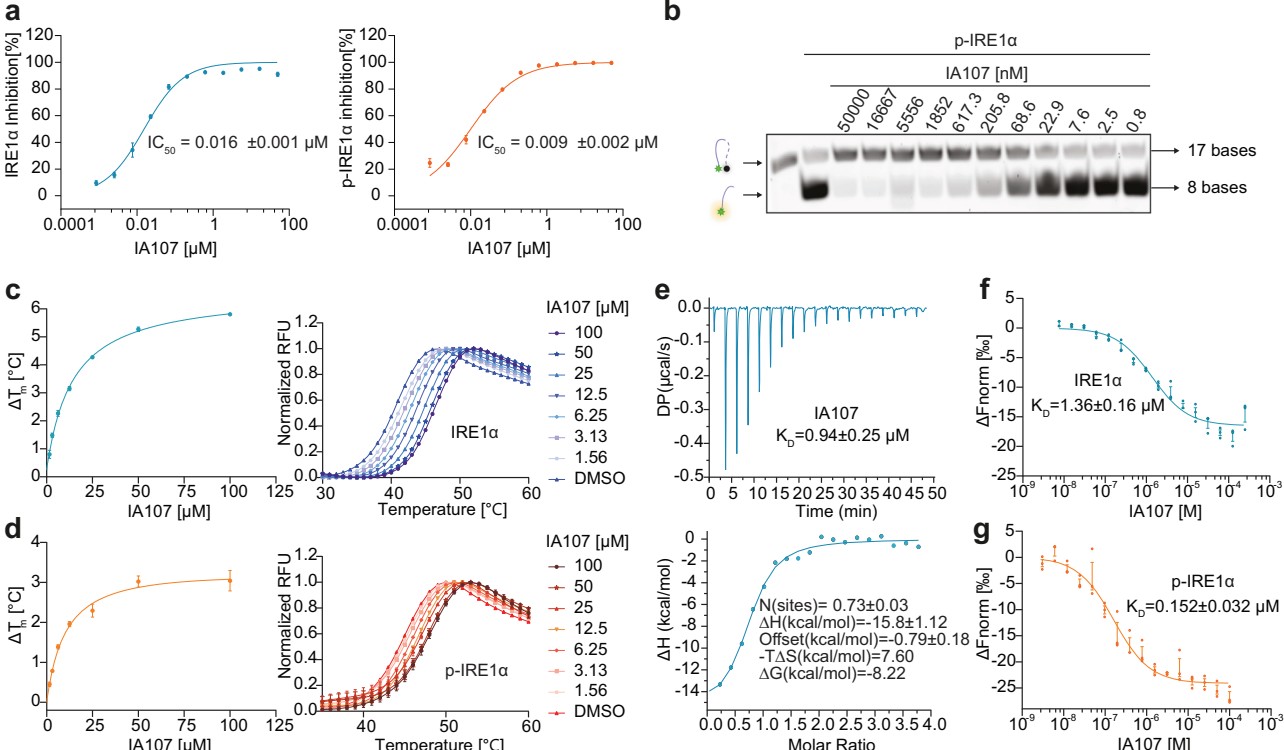

**Fig. 4 | IRE1α Inhibitory activity and binding affinity of IA107. a** IA107 inhibited IRE1α and phosphorylated IRE1α RNase activity in the FRET assay. Data are presented as mean ± SEM, *n* = 3. **b** IA107 concentration-dependently inhibited phosphorylated IRE1α RNase activity in a gel-based RNA-cleavage assay. The expected number of bases was labeled. Representative result of *n* = 3. **c** IA107 concentration-dependently stabilized IRE1α in DSF. Data are presented as mean ± SEM, *n* = 4.

**d** IA107 concentration-dependently stabilized phosphorylated IRE1α in DSF. Data are presented as mean ± SEM, *n* = 4. **e.** IA107 binding affinity towards IRE1α in ITC (by titrating 200 μM compound to 10 μM IRE1α. **f** IA107 binding affinity towards IRE1α in the MST assay. Data are presented as mean ± SEM, *n* = 3. **g** IA107 binding affinity towards p-IRE1α in MST assay. Data are presented as mean ± SEM, *n* = 3. Data are presented as mean ± SEM, *n* = 3.

in the absence of G1749 (Fig. 4e), while the presence of G1749 at 5 μM decreased the binding affinity ($K_D$ 2.25 μM) and IA107 did not show detectable binding to IRE1α in the presence of 50 μM or 200 μM G1749 (Supplementary Fig. 4e). The kinase inhibitory activities of selected IA compounds (IA01, IA06, IA10, IA64, and IA107) against IREα were measured using LanthaScreen™ Eu kinase binding assay (Fig. 5e). Compound IA107 (RNase activity: dephosphorylated IRE1α, $IC_{50}$: 16 nM; p-IRE1α, $IC_{50}$: 9 nM) showed the most potent inhibition among the tested IA compounds with an $IC_{50}$ of 768 nM (kinase activity). Compound IA64 (RNase activity: dephosphorylated IRE1α, $IC_{50}$: 30 nM; p-IRE1α, $IC_{50}$: 30 nM) showed only slightly decreased kinase activity ($IC_{50}$: 1104 nM) in comparison with that of IA107. Compound IA01 (RNase activity: dephosphorylated IRE1α, $IC_{50}$: 300 nM; p-IRE1α, $IC_{50}$: 320 nM) and IA10 (RNase activity: dephosphorylated IRE1α, $IC_{50}$: 1810 nM; p-IRE1α, $IC_{50}$: 1970 nM) showed weaker kinase inhibitory potency with $IC_{50}$ of 2470 nM and 9610 nM, respectively. Compound IA06 (RNase activity: dephosphorylated IRE1α, $IC_{50}$ 7.32 nM; p-IRE1α, $IC_{50}$: 7.79 nM) showed the least active kinase inhibitory activity, with only 23% inhibition at the highest tested concentration of 50 μM. In a word, the kinase inhibition results further confirmed the binding of IA107 to the kinase pocket of IRE1α, together with the observation of a positive correlation between the kinase and RNase activities of the IA compounds.

## IA 107 binding mode study via co-crystal structure

The co-crystal structure of IA107 bound to the phosphorylated IRE1α was successfully resolved in this study, providing valuable mechanistic insights into the inhibition mode for the indole inhibitors at a 3-Å resolution (PDB 9gow, Fig. 6, Supplementary Table S1). The obtained structure contained four copies within the asymmetric unit, chain A

and chain B formed a back-to-back dimer (Fig. 6a), and chain C formed the same dimer with chain D from the molecule in the next asymmetric unit. The density of IA107 was well resolved in all four chains and different orientations of the ligand were tested but only one fit the electron density, indicating the full occupancy of the ligand IA107 in all chains with only one possible conformation.

The 2Fo-Fc density maps of IA107 bound at the ATP-binding pocket of each chain are shown in the Supplementary Fig. 5a. The propionic acid group of IA107 extends towards the pocket proximal to the DFG-motif (Asp-Phe-Gly, residue 711-713), which is a conserved motif located at the beginning region of the kinase activation loop. The N-benzoyl motif of IA107 faces towards the adenine pocket and interacts with the kinase hinge region (Fig. 6b, Supplementary Fig. 5b). Crucial hydrogen bond interactions were formed between the propionic acid of IA107 with the backbone of D711 and with the catalytic K599. The removal or modification of the propionic acid led to a loss of IRE1α inhibitory activity (Fig. 3a) echoing the importance of such hydrogen bond interactions. Another important hydrogen bond was formed between the carbonyl amide of IA107 and the backbone of C645. To validate the importance of this hydrogen bond formation, we performed a structural modification by removing the carbonyl group from the N-benzoyl motif to yield compound IA164, which as expected showed a complete loss of activity (Supplementary Fig. 5c.). Furthermore, a cyclohexane-substituted analogue IA165 was synthesized to evaluate the binding at the hydrophobic channel, which showed that the presence of an aromatic group is preferred as IA165 showed a decreased activity in comparison with that of IA107 (Supplementary Fig. 5d). The cyclohexane and phenyl groups are both hydrophobic and can form hydrophobic interactions with the hinge region, however, the electron-donating property of the cyclohexane group may

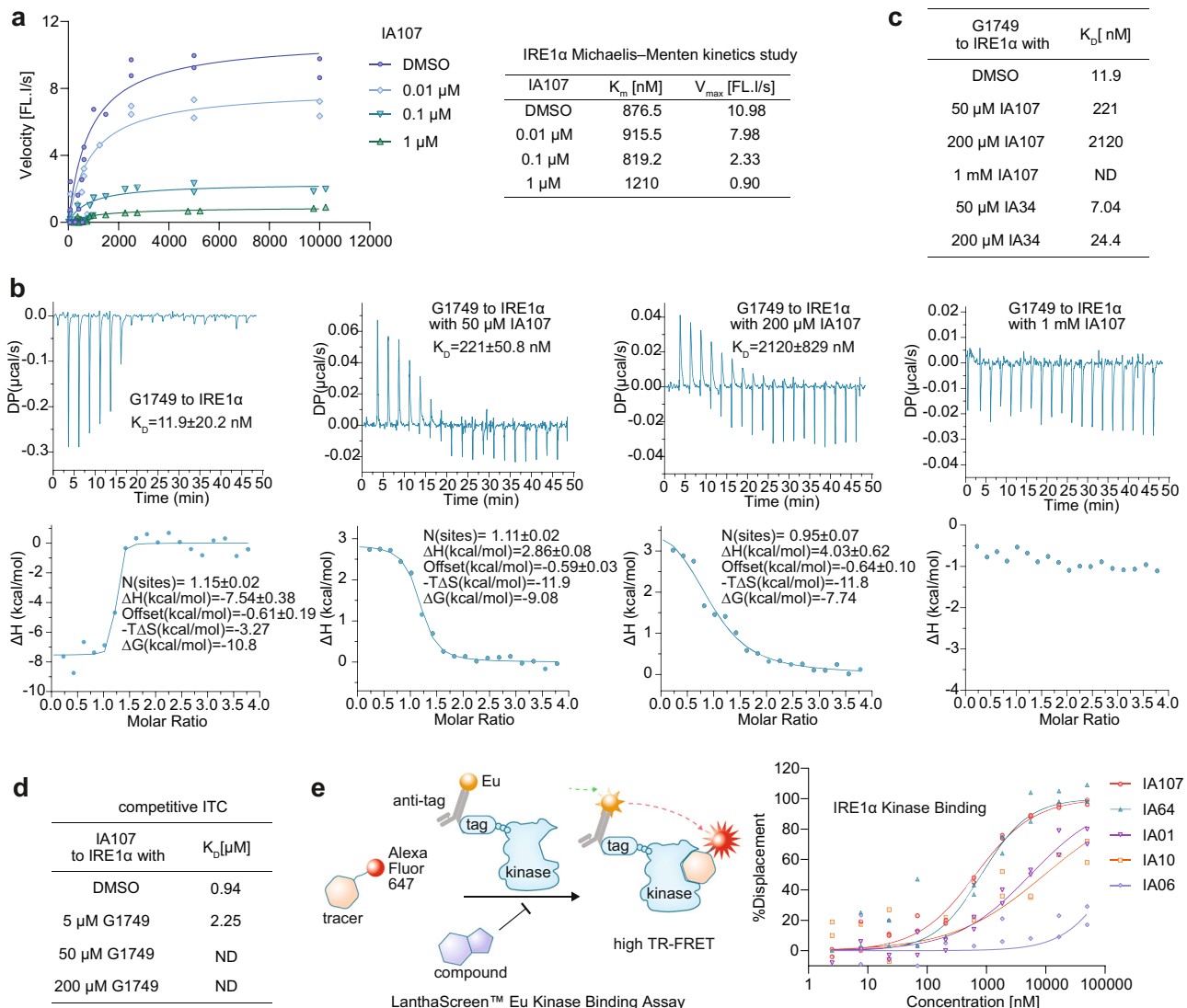

**Fig. 5 | Biophysical characterization of A107 and the inhibitory mechanism studies. a** Michaelis–Menten kinetic study revealed IA107 as a noncompetitive inhibitor of the *XBP1* mRNA. $n = 2$. **b** Competition experiments in ITC by titrating 200 μM G1749 to 10 μM IRE1α pre-incubated with DMSO and different concentrations of IA107 (50 μM, 200 μM, 1 mM), respectively. **c** $K_D$ of the competition experiments in ITC by the titration of 200 μM G1749 to IRE1α pre-incubated with different concentrations of IA107, IA34, or DMSO. **d** Table of $K_D$ values obtained from competition ITC experiments by titrating IA107 to IRE1α pre-incubated with DMSO or different concentrations of G1749. ND, not detectable. **e** The kinase inhibitory activities of IA01, IA06, IA10, IA64, and IA107 against IRE1α in the LanthaScreen™ Eu kinase binding assay (Thermo Fisher SelectScreen), $n = 2$.

weaken the hydrogen bond formation between the amide carbonyl group and C645, leading to the observed decreased activity of IA165. Given that the αC-helix, DFG motif, and the K599-E612 salt bridge in the IRE1α kinase domain are crucial structural elements contributing to the regulation of the RNase domain activity[46], we further evaluated the conformation changes of these structural elements in the IA107-bound p-IRE1α structure in comparison with that in the reported apo-form p-IRE1α structure[42]. IA107 interacts with K599 with the observed N-terminal αC-helix shift upon IA107 binding, and the salt bridge between K599 and E612 of the αC-helix remains intact (Fig. 6b). In contrast, the binding of two reported inhibitors KIRA8 and G0701[42,47], which is a G1749 analogue that showed inhibition activity against p-IRE1, showed that the αC-helix was pushed out together with the breaking of the K599-E612 salt bridge (Supplementary Fig. 5e). The conserved DFG motif does not show any significant shift when aligned with the IA107-binding p-IRE1α structure with the apo form structure, with the DFG-motif adopting a DFG-in conformation in both structures (Supplementary Fig. 5f). The activation loop (residues 711-741) of

IA107-bound p-IRE1α in a slightly shifted conformation but the overall structure is similar to that of the apo-form (Supplementary Fig. 6a). The intact K599 and E612 salt bridge, DFG-in conformation, phosphorylated activation loop and outward pushed N-terminal αC-helix of chain A, C and D were present in Supplementary Fig. 6b–e. The four copies in the asymmetric unit showed similar conformational changes for the evaluated structure elements upon IA107 binding.

IRE1α assembles through dimerization and oligomerization to promote RNase activity[16,18,45]. ATP pocket-binding modulators that interfere with the RNase activity have been reported to activate RNase by promoting IRE1 dimerization and to inhibit RNase by inhibiting dimerization[16,28,38]. In the resolved structure, chains A, B, and C showed good density, while chain D was not very well resolved. As the RNase domain of chain D was not well resolved and the residues K851 to M948 were eliminated due to the poor electron density, we mainly focused on the dimer formed by chain A and chain B in the following analysis and discussion. Compared to the apo-p-IRE1α, the interaction of IA107-bound p-IRE1α kinase domain from two protomer are not significantly

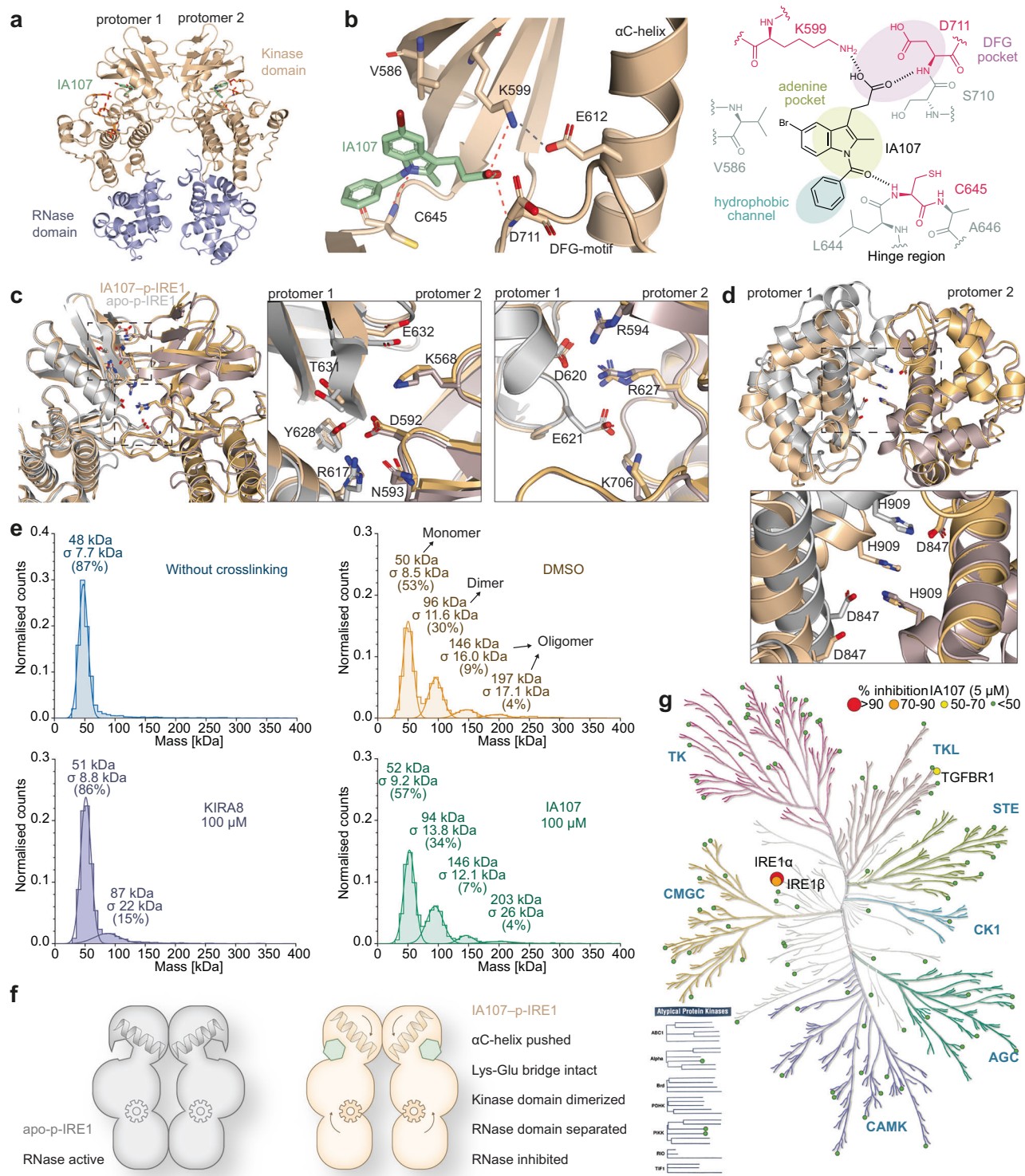

**Fig. 6 | Resolved Structure of IA107-bound p-IRE1α and the selectivity profile of IA107. a** Back-to-back dimer of IA107-bound p-IRE1α (chain A and chain B). **b** Hydrogen bond interactions between IA107 and p-IRE1α. **c** Superimposition view of the kinase dimer interaction from the IA107−p-IRE1α AB dimer (wheat, PDB 9gow) and apo-pIRE1α (gray, PDB 6w3c). The conformations of key residues for the dimer interface interaction in the kinase domain are similar between the apo-form p-IRE1α (gray) and IA107-bound p-IRE1α (wheat). **d** Top-down views of the RNase domain overlay and an enlarged view showing the key interaction residues D847 and H909. **e** IA107 did not inhibit the dimerization of p-IRE1α while KIRA8 did. Measured by Refeyn mass photometry. **f** The graphical summary of the IA107 inhibition mechanism in comparison with that of the apo-p-IRE1. **g** The kinome phylogenetic tree showing the result of IA107 kinase profiling.

affected with a similar overall interaction interface (Supplementary Fig. 7a, b) as well as similar key interface interactions (Fig. 6c). Though a slight conformational shift was observed between the individual residues (Supplementary Fig. 7c), the interaction between the two protomer remains the same, for example, the residue D620, which located at the C-term of the αC-helix, forms hydrogen bonds with R549 and R627 (Fig. 6c). Despite that the interaction of the two protomers of the kinase domain remains the same, the IA107-binding leads to a more

separated conformation between the two RNase domains (835-963) with a reduced overall interaction interface (Fig. 6d, Supplementary Fig. 7b). The hydrogen bonds formed within the RNase domain dimer between H909 and D847 were broken upon binding, further displacing the key catalytic residues H910 and the ribonuclease active site[44], which we proposed is the main reason leading to the inhibition of the IRE1α RNase activity upon IA107 binding (Fig. 6d, Supplementary Fig. 7d). The interaction interface areas of the dimers were analyzed using the PDBePISA web server[48], which showed only a minimal changes up to 1.63% decrease in the kinase domain interface in comparison with apo form dimer. For the RNase domain of dimer AB, the interaction interface decreased by ~26.2% (Supplementary Fig. 7e). We assumed that the key interactions between two kinase domain protomers would be sufficient to maintain the dimerization of p-IRE1α upon IA107 binding. To evaluate the effect of IA107 against IRE1α dimerization and oligomerization, a DSS (disuccinimidyl suberate) cross-linking assay with mass photometry measurement was performed using p-IRE1α. The reported inhibitor KIRA8 and G1749, which showed inhibitory activity against p-IRE1α RNase activity and dimerization, were used as reference compounds[20,42,47,49]. The results showed that different from KIRA8 and G1749 interfere with the dimerization and inhibit the p-IRE1α RNase activity, compound IA107 did not inhibit the dimerization and oligomerization of p-IRE1α (Fig. 6e and Supplementary Fig. 7f, g), which confirmed our assumption. Overall, unlike most reported kinase domain-binding and RNase domain-inhibiting molecules, which inhibit the RNase activity by inhibiting the dimer formation, IA107 binds to the kinase pocket of IRE1α with the pushed-out αC-helix and the presence of the intact K599-E612 salt bridge. The kinase domains of the two protomers interact with each other in the back-to-back dimeric conformation upon IA107 binding, maintaining the dimeric and oligomeric assemblies of p-IRE1α but the RNase domains are more separated, resulting in the inhibition of the RNase activity of IRE1α (Fig. 6f).

Overall, the resolved complex structure between p-IRE1α and IA107 further supports the allosteric inhibition mechanism of IA107, in line with the non-competitive inhibition mode against the *XBP1* RNA substrate. While crystal packing may cause small shifts of the side chains, the structure of the IA107-bound p-IRE1α complex suggests the ligand-induced conformational change upon binding to the IRE1α kinase site, which matched the allosteric inhibition mode suggested by the biochemical and biophysical data.

## IA107 is selective against IRE1α over other kinases

Upon the clarification of its inhibitory mechanism, we next evaluated the selectivity of IA107 in a kinase profiling assay against a selection of kinase targets (Fig. 6g, Supplementary Table S2). Overall, compound IA107 showed minimal (<10%) to low (<30%) inhibitory activity against the evaluated kinases at a testing concentration of 5 μM. In addition to IRE1α (ERN1), IA107 inhibited the kinase domain of the IRE1β (ERN2) isoform with an IC$_{50}$ of 1440 nM in the LanthaScreen™ Eu kinase binding assay (Supplementary Fig. 7h). The observed 2-fold selectivity for IRE1α over IRE1β is not surprising given the 80% identity in the kinase domains between the IRE1α and IRE1β isoforms[50]. Except for IRE1α and IRE1β, the only kinase that showed more than 50% inhibition in the kinase profiling is TGFBR1 (54%).

## IA107 inhibited ER stress and induced *XBP1* mRNA splicing

Considering that IRE1α activation triggered by cellular ER stress induces *XBP1* mRNA splicing[21], we continued with the evaluation of the downstream effects induced by IA107 treatment in A549 cells. We pre-treated the cells with IA107 and induced the ER stress by tunicamycin (Tm). The spliced *XBP1* mRNA level was then evaluated by RT-qPCR. As shown in Fig. 6a, tunicamycin induced the *XBP1* splicing, while the pretreatment with IA107 concentration-dependently inhibited the *XBP1* mRNA splicing. The binding of IA107 to the wild-type full-length

IRE1α was evaluated by a thermal stability assay performed using A549 cell lysate with western blot analysis (Supplementary Fig. 8a). The result showed that IA107 stabilized the full-length IRE1α in a concentration-dependent manner, indicating the binding of IA107 to the wild-type IRE1α. However, the inhibition on the cellular *XBP1* splicing upon IA107 treatment was weaker in comparison with the inhibitory results shown in in vitro assays, which might be because of the limited cell membrane permeability of IA107 imposed by the liable groups such as the propanoic acid. Whereas the carboxylic acid-containing compounds usually showed poor cellular permeability[51], it has been demonstrated that the propanoic acid group is crucial for the inhibitory potency of the indole-based IRE1α inhibitor due to the formation of key hydrogen bond interactions with the catalytic residue (K599) and the asparagine residue (D711) of the DFG-motif. In this context, to overcome the permeability liability and at the same time to maintain the crucial function of the carboxylic acid, we resorted to a prodrug strategy by using an ester-containing analogue IAPD1 (Fig. 7b). IAPD1 did not show any detectable activity against IRE1α in the FRET assay and LanthaScreen kinase binding assay as expected (Supplementary Fig. 8b, c). The treatment of IAPD1 inhibited the ER stress-induced *XBP1* mRNA splicing with a cellular IC$_{50}$ of 180 nM (Fig. 7c).

In parallel, the *IRE1α* transcription level was not affected by the IAPD1 treatment (Supplementary Fig. 8d). The activity of IAPD1 against the ER stress-induced *XBP1* mRNA splicing, and the *IRE1α* transcription level were further evaluated in HCT 116 and HT-29 cell lines (Supplementary Fig. 8e–h). Consistent results were observed in all three tested cell lines. IAPD1 treatment inhibited the *XBP1* mRNA splicing with IC$_{50}$ of 220 nM and 160 nM in HCT 116 cells and HT-29 cells, respectively (Supplementary Fig. 8e, g), with no obvious impact on the transcription level of *IRE1α* (Supplementary Fig. 8f, h).

The translated XBP1s protein and IRE1α protein levels were evaluated via western blot in both A549 and MDA-MB-231 cells (Fig. 7d, Supplementary Fig. 8i). KIRA8 and G1749 were used as reference compounds[42]. Under tunicamycin stimulation, the XBP1 protein concentration increased in both cell lines. The pre-treatment of IA107 and KIRA8 inhibited the ER stress-induced XBP1s protein increment. Meanwhile, treatment of G1749 at either 1 μM or 5 μM partially inhibited the XBP1 protein induced by ER stress. The treatment of compounds IAPD1, KIRA8, and G1749 did not affect the IRE1α protein level, indicating that the observed activity of the compounds against IRE1α downstream proteins was not due to the effect on the IRE1α protein as degraders. Further target engagement of IAPD1 was confirmed by the cellular thermal shift assay in A549 cells, which demonstrated that the thermal stability of cellular IRE1α was concentration-dependently increased in comparison to that of the DMSO control upon IAPD1 treatment, with the thermal stability of the housekeeping protein GAPDH remained unaltered (Fig. 7e). Moreover, the antiproliferation activity of IA107 and IAPD1 were evaluated in A549, MDA-MB-468, HCT 116, and HT-29 cells (Supplementary Fig. 8j). Both compounds showed negligible effects in the evaluated cells with less than 30% inhibition at the highest tested concentration of 50 μM, suggesting that IA107 and IAPD1 are not cytotoxic.

In summary, IRE1α is an important UPR sensor with dual kinase and ribonuclease functions that regulate ER stress and the downstream UPR signaling pathways, dysregulation of which has been associated with various human diseases. Given the therapeutic potential of IRE1α-targeting small molecules and the scarcity of such chemotypes, in this study, we identified a series of substituted indoles as an IRE1α-inhibiting chemotype with excellent potency and selectivity. Starting from an initial screening hit, we investigated the scope of the structural modifications surrounding the indole scaffold, which revealed that the propanoic acid group at the C-3(R$^1$) position is crucial and the indole nitrogen position (R$^2$) and the C-5(R$^3$) position were tolerated for certain substitution groups. The most potent inhibitor IA107 showed IC$_{50}$ of 16 nM and 9 nM towards dephosphorylated and

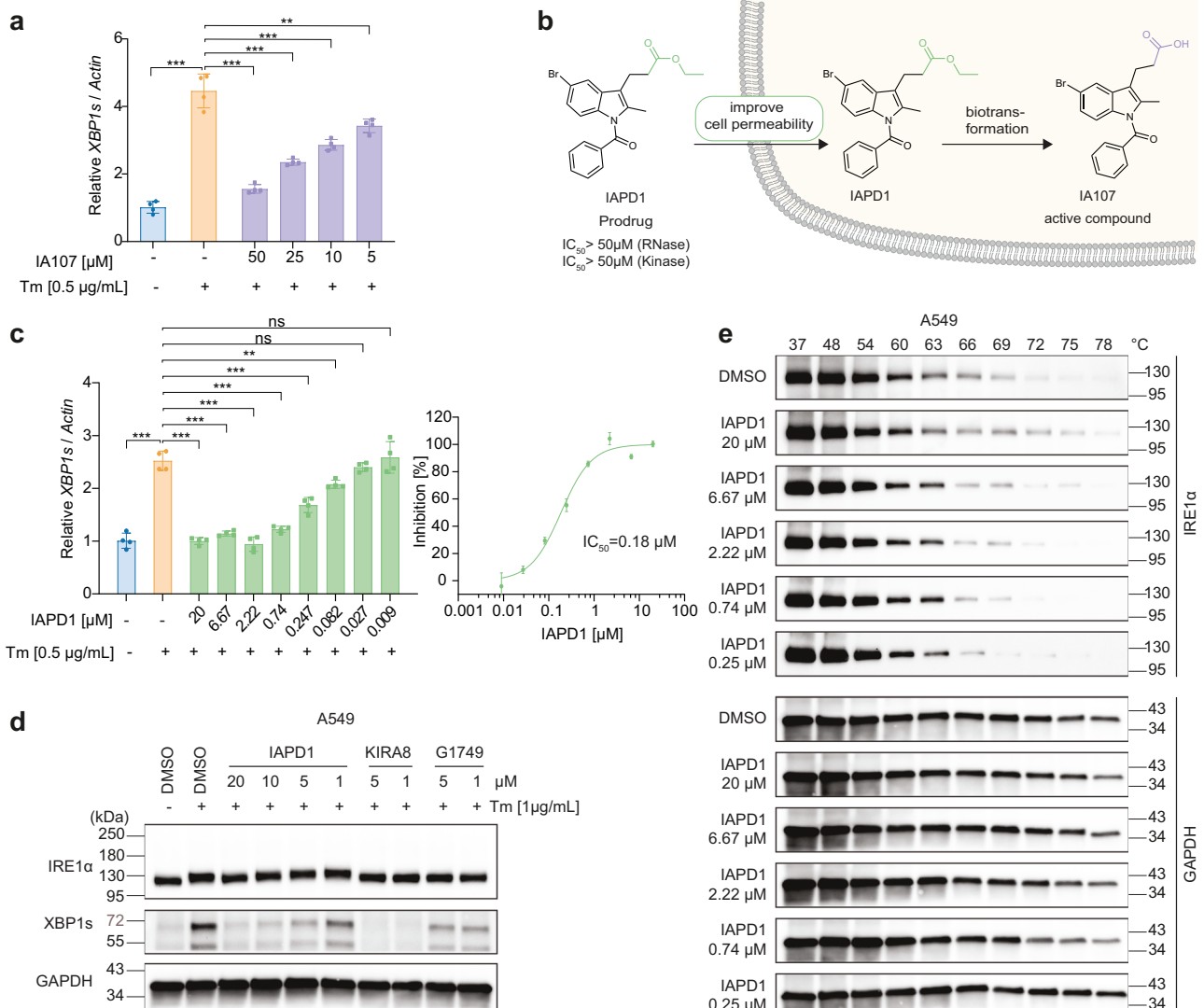

**Fig. 7 | Target engagement of IA107 and the prodrug strategy using IAPD1 to improve the cellular uptake. a** IA107 concentration-dependently inhibited ER stress-induced *XBP1* splicing in A549 cells with decreased activity upon pre-treatment with IA107 for 2 h and co-treatment with tunicamycin (Tm) for 2 h. Data are presented as mean ± SEM, $n = 4$. The $p$ values were calculated using a one-tailed independent Student's $t$ test, **$p < 0.01$, ***$p < 0.001$. The exact $p$ values were provided in the Source data file. **b** A prodrug strategy to improve cellular permeability by employing an ester prodrug IAPD1. **c** IAPD1 inhibited ER stress-induced *XBP1* splicing in A549 cells with an IC$_{50}$ of 0.18 μM. Data are presented as

mean ± SEM, $n = 4$. The $p$ values were calculated using a one-tailed independent Student's $t$ test, **$p < 0.01$, ***$p < 0.001$, ns $p$å 0.05. The exact $p$ values were provided in the Source data file. **d** IAPD1 inhibited the XPB1s protein level in A549 cells in the Western blot assay. Cells were treated with indicated compounds for 2 h, then a final concentration of 1 μg/mL tunicamycin was added into the cells and incubated for another 4 h before cells were lysed, $n = 3$. **e** Cellular thermal shift assay showing IAPD1 concentration-dependently stabilized IRE1α in A549 cells upon the treatment with different concentrations of IAPD1 or DMSO for 4 h, $n = 3$.

phosphorylated IRE1α, binding affinity was confirmed by orthogonal assays including MST (IRE1α K$_D$: 1.36 μM, p-IRE1α K$_D$: 152 nM) and ITC (K$_D$: 0.94 μM). IA107 may preferentially bind to the active dimeric proteins, which could cause the different affinities p-IRE1α and IRE1α and the observed discrepancy between the binding affinity and functional inhibitory potency. The discrepancy may also be attributed to the allosteric mechanism of IA107, which binds to the kinase domain and induces conformational changes to inhibit the RNase activity rather than directly competing with the RNA substrate. After the dissociation of IA107, IRE1α may remain in the inactive conformation, contributing to the sustained inhibition effect. The resolved IA107–p-IRE1α complex structure revealed the inhibitory mechanism of IA107, which functioned as a kinase domain binder that allosterically inhibited the IRE1α RNase activity. The binding of IA107 did not impact the formation of the Lys–Glu bridge and the high-order assembly of IRE1α, instead, IA107 inhibited the RNase activity by separating the RNase

domain dimer interface, confirming the formation of the RNase dimer is critical for the IRE1 RNase activity. Furthermore, IA107 concentration-dependently inhibited the ER stress-induced *XBP1* splicing in cells, for which the prodrug IAPD1 showed a 50-fold improvement of the cellular activity in inhibiting the *XBP1* mRNA splicing. The low antiproliferation activities indicated that the characterized IRE1α inhibitors are not cytotoxic and could be useful to be studied in human diseases beyond cancers, such as pain[14,52]. Given that the IA compound did not inhibit the dimerization and oligomerization of the IRE1α cytoplasmic domain in the in vitro assay, we expect that the IA compound would not inhibit the assembly in the cellular context either, since the luminal domain drives the IRE1α dimerization in cells. Therefore, the impact of IA compounds on IRE1α assembly in a cellular context warrants further investigation to fully understand the inhibitory mechanism. Collectively, our results established indole-containing compounds as an IRE1α-inhibiting chemotype with

excellent potency and selectivity, which expanded the potential biological application scope associated with IRE1α-targeting small molecules.

## Methods

### Protein expression and purification

Human IRE1α kinase and RNase domain (residues 547–977) was subcloned into pLIB plasmid with a His tag and TEV cleavage site, and expressed using Bac-to-Bac expression system. The primers used are listed in Supplementary Table S3. The protein purification was performed following established procedures[42]. Briefly, Sf9 cells were infected with baculovirus and incubated at 27 °C 110 rpm for 72 h. The cells were collected by centrifugation at $4000 \times g$ for 15 min and the cell pellet was resuspended in lysis buffer (50 mM HEPES pH 7.5, 300 mM NaCl, 10% glycerol, 1 mM MgCl$_2$, 1 mM TCEP, 5 mM imidazole), then supplemented with 1:10,000 benzonase (Sigma, E1014), 1 tablet/ 25 mL SIGMAFAST™ Protease Inhibitor Cocktail tablets (Sigma, S8830) and 1 mM PMSF before lysed by sonication at 4 °C. Then sample was centrifuged at $48,000 \times g$ for 45 min, the supernatant was filtered and loaded onto a nickel-affinity column (GE Healthcare, Ni Sepharose™ 6 Fast Flow) pre-equilibrated with lysis buffer, the column was washed by lysis buffer containing 30 mM imidazole and the IRE1α protein was eluted in lysis buffer containing 300 mM imidazole. The protein elution was incubated with TEV protease to remove His tag at 4 °C overnight. Then the sample was diluted 1:6 in buffer (50 mM HEPES pH 7.5, 1 mM TCEP, 5% glycerol) and loaded onto a 5 mL pre-packaged Q HP anion exchange column (Cytiva, HiTrap Q HP) pre-equilibrated with QA buffer (50 mM HEPES pH 7.5, 50 mM NaCl, 1 mM TCEP, 5% glycerol). The unphosphorylated and phosphorylated protein was separated by eluting the protein with a very shallow gradient with 0-30% buffer QB (50 mM HEPES pH 7.5, 1 M NaCl, 1 mM TCEP, 5% glycerol) over 80 CV. Different peaks were collected separately and verified by LC-MS. The fully phosphorylated fraction was collected and further purified by size exclusion chromatography (Cytiva, HiLoad 16/600 Superdex 200 pg,) with SEC buffer (25 mM HEPES pH 7.5, 250 mM NaCl, 1 mM TCEP, 10% glycerol) and fractioned as monomeric peak. The rest protein fraction from the Q HP anion exchange column was incubated with Lambda protein phosphatase (New England BioLabs, P0753S) overnight at 4 °C, the dephosphorylation was confirmed by LC-MS. The unphosphorylated protein was further purified by size exclusion chromatography with the SEC buffer described above. The purified phosphorylated (p-IRE1α) and dephosphorylated (IRE1α) protein aliquots were stored at −80 °C for future usage.

### Protein mass spectrometry

IRE1α proteins were diluted in PBS buffer for LC-MS. The protein LC-MS was performed using an Agilent 1260 II Infinity system equipped with an electrospray ion source in positive mode, run through a desalting cartridge (AdvanceBio Desalting-RP, 2.1 mm, 12.5 mm, Agilent) with a gradient of 5–80% HPLC-grade acetonitrile (supplemented with 0.1% TFA) in HPLC-grade water (supplemented with 0.1% TFA) (flowrate: 0.4 mL/min, runtime: 6 min). The spectra were deconvoluted using the ProMass software (Novatia).

### FRET-based IRE1α RNase activity assay

The dual-labeled RNA substrate with a FAM at 5' and the BHQ-1 at 3' containing the *XBP1* sequence (5'FAM-CAUGUCCGCAGCGCAUG-3' BHQ1) was purchased from Integrated DNA Technologies (IDT). The FRET-based IRE1α RNase activity assay was carried out in reaction buffer containing 20 mM HEPES pH 7.5, 50 mM KOAc, 1 mM MgOAc$_2$, 1 mM DTT, 0.05% v/v TritonX-100, at a total reaction volume of 20 μL in a 384-well black plate (4514, Corning). For unphosphorylated IRE1α (IRE1α), a final concentration of 40 nM IRE1α protein and 100 nM RNA substrate was used. For phosphorylated IRE1 (p-IRE1α), a final concentration of 4 nM p-IRE1α protein and 200 nM RNA substrate was

used. The indicated compound and IRE1α protein were incubated for 30 min at room temperature before the substrate was added to the reaction. The plate was then kinetically read over one hour at room temperature using a TECAN Spark plate reader. The measurement was carried out using the fluorescence top reading mode at the excitation wavelength of 485 nm and the emission wavelength of 535 nm with a 10 nm excitation and emission bandwidth. The linear part of the reaction curve was adopted and fitted with sample linear regression to obtain the slope, and the slope was adopted to reflect the reaction rate. The inhibition % was calculated by using the equation: inhibition % = 100 × (slope of DMSO control–slope of the sample)/(slope of DMSO control–slope of blank control), DMSO control: DMSO with protein and RNA substrate; Blank control: DMSO with RNA substrate. The IC$_{50}$ values were calculated using the concentration-response model in GraphPad Prism 9.

### Compound screening

The small-molecule screening was performed against an in-house chemical library containing 11393 compounds provided by the Compound Management and Screening Center (COMAS), MPI Dortmund, by using the FRET assay described above with 5 μM compound, 40 nM IRE1α, 100 nM dual labeled RNA substrate in 384-well black plates (4514, Corning). The detailed small-molecule screening information is shown in Supplementary Table S4.

### Gel-based RNA-cleavage assay

Purified p-IRE1α with a final concentration of 10 nM was preincubated with the indicated compounds for 30 min at room temperature, then 50 nM FAM and BHQ-labeled *XBP1* substrate were added to the reaction. The reaction was further incubated for 45 min at room temperature and then mixed with 6× TBE loading dye (45% H$_2$O, 40 % glycerol, 15% 10× TBE, 0.1% bromphenol blue). 5 μL sample was loaded to 15% pre-runed urea-PAGE, and the gel was run in pre-warmed (-60 °C) 1× TBE buffer buffer in the dark. The gel fluorescence was detected and imaged using the ChemiDoc MP imaging system (Bio-Rad).

### Differential scanning fluorimetry (DSF) assay

The DSF assay was performed in PBS buffer containing 2 mM DTT, in a total reaction volume of 20 μl with indicated compounds (final 1% DMSO) and the final concentration of 1 μM IRE1α protein and 5× SYPRO Orange fluorescent dye (5000× in DMSO, Sigma S5692). The melting curve was measured at a temperature range from 25 °C to 95 °C and in increments of 1 °C for 30 s using a Bio-Rad CFX96 Real-Time PCR detection system with the FRET scan mode. The melting temperature (T$_m$) was obtained by fitting the melting curve to Boltzmann sigmoidal in GraphPad Prism and the thermal shift (ΔT$_m$) was calculated by using the equation: $\Delta T_m = T_{m(compound)} - T_{m(DMSO)}$.

### Isothermal titration calorimetry (ITC) assay

The dephosphorylated IRE1α protein was used directly after purification, ITC experiments were carried out at 25 °C using MicroCal PEAQ-ITC system (Malvern) in buffer 25 mM HEPES pH 7.5, 250 mM NaCl, 1 mM TCEP, 5% glycerol, 1% DMSO. The solution was diluted to a final 1% DMSO in the same buffer as for the protein. Protein and compound solution were degassed before loading to the cell and syringe respectively. Unless otherwise noted, 200 μM of compound was loaded into the syringe and 10 μM of IRE1α protein was loaded into the sample cell. For competitive ITC, the protein was incubated with the indicated compound at 25 °C (final 1% DMSO) before loading to the cell. Instrument settings were: reference power 10 μcal/s, feedback high, stir speed 750 rpm, initial delays 60 s, injection spacing 150 s, injection duration 4 s. All experiments were performed with a single 0.4 μL injection followed by 18 (or 15), 2 μL injections. Data were analyzed using MicroCal PEAQ-ITC analysis software. The one set of sites

fitting model is used for the calorimetry data analysis, with cell concentration and syringe concentration as the input parameters. For compound IA01, the N was set to 1 to improve the fit.

## Microscale thermophoresis (MST)

Dephosphorylated IRE1α protein and phosphorylated IRE1α protein were labeled and purified using Protein Labeling Kit RED-NHS 2nd Generation (NanoTemper, MO-L011), following the manufacturer's protocol. The labeling buffer contained 50 mM HEPES, pH 7.5, 200 mM NaCl, 2% glycerol, 0.5 mM TCEP, 0.05% tween20. MST experiments were performed using the Monolith Nt.115 instrument (NanoTemper), and binding experiments were conducted according to the instrument's instructions. Compounds were serially diluted 2-fold and incubated with labeled IRE1α protein at a final concentration of 20 nM. The highest compound concentration measured was 250 μM with a final DMSO concentration of 2.5%. Measurements were carried out using the red detector with 60% excitation power. Data were analyzed using the manufacturer's software, and results from three individual measurements were merged for analysis.

## LanthaScreen™ Eu kinase binding assay

LanthaScreen™ Eu kinase binding assays were conducted by SelectScreen Kinase Profiling Services (Thermo Fisher) using Alexa Fluor conjugated kinase tracer, tagged kinase protein, and Eu-labeled anti-tag antibody. The binding of the tracer to the kinase will form the tracer-kinase complex, resulting in a high FRET signal with the Eu-labeled anti-tag antibody. Displacement of the tracer by a kinase inhibitor led to a decreased FRET signal. For the IRE1α (ERN1) kinase binding assay, 5 nM of ERN1, 2 nM of Eu-anti-GST, and 100 nM tracer 236 (with Kd = 160 nM) was used. For the IRE1β (ERN2) kinase binding assay, 5 nM of ERN2, 2 nM of Eu-anti-GST, and 100 nM tracer 236 (with Kd = 108 nM) was used. Buffer A containing 50 mM HEPES pH 7.5, 0.01% BRIJ-35, 10 mM MgCl₂, and 1 mM EGTA was used.

## Co-crystallization of IA107-pIRE1α

The purified p-IRE1α protein (15 mg/mL in buffer containing 25 mM HEPES pH 7.5, 150 mM NaCl, 1 mM TCEP) was incubated with a final concentration of 1 mM IA107 (with a final concentration of 1% DMSO) at 4 °C overnight. The protein-ligand mixture was centrifuged at 18,000 × g for 30 min at 4 °C before crystallization setup. The reservoir solution contains 0.05 M bicine pH 9.0, 36% v/v PEG300, and 0.12 M NaCl. Crystallization was performed by mixing 1 μL of reservoir buffer into 1.5 μL of the protein-ligand mixture in a hanging drop setup using 24-well plates. Thin plate-shaped crystals grew over 3-4 days at 20 °C. Crystals were fished from the drop and flash-frozen in liquid nitrogen. Synchrotron data was collected from the ID30A-3 beamline at the European Synchrotron Radiation Facility (ESRF). The data was processed and scaled using XDS and XSCALE[53]. The structures were solved by molecular replacement using PHASER[54] within the PHENIX software suite[55]. The topology file for ligand IA107 was generated using AceDRG[56] within the CCP4 suite[57], based on its SMILES string. The structure was manually refined using COOT[58] and refined using phenix.refine[59] program. Part of chain D (K851 to M948) was eliminated due to the poor electron density data.

## DSS-crosslinking

Phosphorylated IRE1α (p-IRE1α) and compounds were incubated 30 min at room temperature, with a final concentration of 2 μM p-IRE1α and 100 μM of the indicated compound, in a buffer containing 20 mM HEPES, pH 7.5, 50 mM KOAc, 1 mM MgOAc₂, 1 mM DTT, 0.05% v/v TritonX-100, in a total reaction volume of 20 μL. After incubation, DSS (disuccinimidyl suberate, Thermo Fisher, A39267) was added to the reaction at a final concentration of 250 μM to induce crosslinking at room temperature for 1 h. Tris (pH 7.5) was added to a final concentration of 50 mM to stop the reaction. Samples were further

measured with a mass photometry instrument (Refeyn Ltd). The PBS buffer was used for the mass photometry measurement. The instrument was calibrated with BSA and TG protein mixture in PBS. Samples were diluted to a protein concentration of 100 nM with buffer before measurement. Then, 18 μL PBS buffer was added to the center of the gasket on the coverslips to check and find the focus. Afterward, 2 μL of the diluted sample was added, mixed, and measured. Data were analyzed using Discover MP software, for each sample, results from 3 individual measurements were merged.

## Kinase profiling

Kinase profiling was conducted by the SelectScreen Kinase Profiling Services (Thermo Fisher), using Z'-LYTE, Adapta, or LanthaScreen Eu Kinase Binding Assay for different kinases. Targets, assays, and used ATP concentrations are listed in Table S2. The kinome phylogenetic tree was generated using the KinMap web portal.

## Cell culture

A549, HT-29, and MDA-MB-468 cell lines were purchased from ATCC (American Type Culture Collection, CCL-185, HTB-38, HTB-132). HCT 116 and MDA-MB-231 cell lines were purchased from DSMZ (German Collection of Microorganisms and Cell Cultures, ACC581, ACC732). The cell lines were cultured in high glucose DMEM medium (Gibco, 61965026) with 10% FBS (Gibco, 10500064,) and 1% penicillin-streptomycin (Gibco, 15140122). All cell lines were cultured at 37 °C with a humidified 5% CO₂ atmosphere.

## Cell viability assay

Cells were collected and seeded into 96-well plates at a density of 2000–4000 cells per well and cultured overnight to allow cells to adhere. After adherence, cells were treated with the indicated compounds for 72 h. Then 20 μL CCK-8 solution (Vazyme, A311) was added into each well and incubated at 37 °C for 2–4 h. Absorbance was detected at a wavelength of 450 nm using a TECAN plate reader. Cell viability was calculated with the equation: Cell viability (%) = 100(Absorbance of treated cell - Absorbance of Blank control)/(Absorbance of DMSO-treated cells - Absorbance of Blank control); Blank control: medium and CCK-8 solution, no cells.

## RT-qPCR

A549 cells, HCT 116 cells, and HT-29 cells were collected and seeded in 6-well plates at a density of $5 \times 10^5$ cells per well and cultured overnight to attach, then treated with compound or DMSO for 2 h, then 0.5 μg/mL final concentration of tunicamycin (Tm, MP Biomedi, 0215002805) were added to induce the ER stress for 2 h. After treatment, total RNA was purified using an RNeasy mini kit (Qiagen, 74106). 500 ng of total RNA was used for reverse transcription using the High-Capacity cDNA Reverse Transcription Kits (Thermo Fisher, 4368814). cDNA was diluted 1: 5 using RNase-free water, and 1 μL CDNA was used for each qPCR reaction. The qPCR was performed using PowerUp™ SYBR™ Green Master Mix (Thermo Fisher, A25742) with 10 μL reaction volume using Bio-Rad CFX96 Real-Time PCR Detection System following the standard cycling mode (primer $T_m \geq 60$ °C) on the manufacturer's protocol. Data was analyzed using the $2^{-\Delta\Delta Ct}$ method. The primers used are listed in Supplementary Table S3. The inhibition % was calculated by using the equation: inhibition% = 100× (normalized Tm control - normalized of the sample)/(normalized Tm control - normalized DMSO control). Tm control: Cells treated with tunicamycin; DMSO control: Cells treated with DMSO only.

## Cellular thermal shift assay (CETSA)

A549 cells were harvested and seeded into 100 mm standard cell culture dishes at 80% confluency and cultured overnight. For the IA107 experiment, cells were gently washed with pre-cooled DPBS (Gibco,14190169) and lysed in RIPA buffer (Sigma, R0278)

supplemented with phosphatase inhibitor (Roch, 4906845001) and 1× protease inhibitor cocktail (Sigma, P8340) for 30 min on ice. Cell lysates were centrifuged at $18,000 \times g$ for 30 min at 4 °C and the supernatants were collected. The lysates were collected together and then aliquoted into 550 µL, the aliquots were incubated with different concentrations of IA107 with a final concentration of 0.1% DMSO for 30 min at room temperature. After the incubation, the sample was aliquoted into 10 PCR tubes, with 50 µL of lysate per tube and further heated at the indicated temperatures using a PCR cycler (25 °C for 1 min; indicated temperatures for 3 min; 16 °C, hold). After heating, the lysates were centrifuged at $18,000 \times g$ for 30 min at 4 °C and the supernatants were collected and mixed with 4x protein loading buffer (Invitrogen, 2463559), heated at 95 °C for 10 min and separated by 4–20% MP stain-free gels (Bio-Rad, 4568095) then transferred to 0.45 µm PVDF membranes (Bio-Rad, 1704274) using the Trans-Blot Turbo Transfer System with the preprogrammed standard SD protocol (Bio-Rad). Membranes were blocked with 5% skim milk (Roth, T145.2) in PBST buffer, then incubated with primary antibody IRE1α (Cell Signaling Technology, #3294 s, 1:1000), GAPDH (Proteintech, 10494-1-AP, 1:6000) overnight at 4 °C with shaking. Membranes were washed with PBST and incubated with the Goat Anti-Rabbit (Proteintech, SA00001-2, 1:6000) at room temperature for 1 h. After washed with PBST, the membranes were visualized using Amersham ECL prime western blotting detection kit (Cytiva, RPN2232) and imaged with a ChemiDoc MP imaging system (Bio-Rad).

For the IAPD1 experiment, A549 cells were treated with different concentrations of IAPD1 or DMSO for 4 h (final 0.1% DMSO). After treatment, cells were gently washed with pre-cooled DPBS (Gibco,14190169) and lysed in RIPA buffer (Sigma, R0278) supplemented with phosphatase inhibitor (Roch, 4906845001) and 1x protease inhibitor cocktail (Sigma, P8340) on ice for 30 min. Lysates were collected by scraping and centrifuged at $18,000 \times g$ for 30 min at 4 °C, and the protein concentrations of the supernatants were quantified using a BCA protein assay kit (Thermo Scientific, 23227) and diluted to 4 µg/µL, then cell lysate was aliquoted into 10 PCR tubes, with 50 µL of lysate per tube. Cell lysate aliquots were further heated at specified temperatures using a PCR cycler (25 °C for 1 min; X °C for 3 min; 16 °C, hold). After heating, samples were centrifuged at $18,000 \times g$ for 30 min at 4 °C, and supernatants were collected and analyzed by western blot using the method and antibodies aforementioned.

### Western blot analysis of IRE1α and XBP1s protein levels

A549 cells and MDA-MB-231 Cells were harvested and seeded into 6-well plates with a density of $8 \times 10^5$ cells per well and cultured overnight. After the cells were attached, the cells were treated with IAPD1, KIRA8, and G1749 with a final concentration of 0.1% DMSO for 2 h, then a final concentration of 1 µg/mL tunicamycin (Tm, MP Biomedi, 0215002805) were added into each well to induce the ER stress for 4 h. After treatment, the cells were washed with pre-cooled DPBS (Gibco,14190169) and lysed by RIPA buffer (Sigma, R0278) supplemented with phosphatase inhibitor (Roch, 4906845001) and 1x protease inhibitor cocktail (Sigma, P8340) on ice for 30 min. Cell lysates were centrifuged at $15,000 \times g$ for 10 min, and the supernatants were collected. Protein concentrations were measured using a BCA protein assay kit (Thermo Scientific, 23227) and all samples were diluted to the same protein concentration. Each sample with 40 µg of total protein was loaded onto a 4–15% MP stain-free gel (Bio-Rad, 4568085) and separated in tris-glycine-SDS running buffer. The proteins were transferred onto a 0.45 µm PVDF membrane (Bio-Rad, 1704274) using the Trans-Blot Turbo Transfer System with the preprogrammed standard SD protocol (up to 1.0 A; 25 V; 30 min). Membranes were blocked with 5% skim milk (Roth, T145.2) in PBST buffer, then incubated with primary antibody IRE1α (Cell Signaling Technology, #3294 s, 1:1000), XBP1s (Cell Signaling Technology, #12782 s, 1:1000) GAPDH (Proteintech, 10494-1-AP, 1:6000) overnight at 4 °C with shaking. After

incubation, membranes were washed with PBST and incubated with the Goat Anti-Rabbit (Proteintech, SA00001-2, 1:6000) antibody at room temperature for 1 h. After washing with PBST, the membranes were visualized using Amersham ECL Prime Western Blotting Detection Kit (Cytiva, RPN2232) and imaged with a ChemiDoc MP imaging system (Bio-Rad).

### Reporting summary
Further information on research design is available in the Nature Portfolio Reporting Summary linked to this article.

## Data availability
Data supporting the findings of the study are available from the corresponding author upon request. The protein X-ray crystal data generated in this study have been deposited in the PDB database under accession code 9GOW. Compound characterizations are provided in the Supplementary Information. All other data are included in the manuscript or in the supplementary Information. Source data for uncropped gels and blots, biochemical experiments, and biophysical experiments are provided with this paper. Previously deposited PDB structures 4PL3, 6W3C, 6W39, 6W3E, 6URC were used for modeling, comparison, and alignment. The IRE1α protein sequence is available through UniProt under the code O75460. Source data are provided with this paper.

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

## Acknowledgements

AstraZeneca, Merck KGaA, Pfizer Inc., and the Max Planck Society are gratefully acknowledged for financial support. The research work in the

Wu group is supported by the Chemical Genomics Centre III (to P.W.), the Boehringer Ingelheim Foundation (Plus 3 Grant to P.W.), the Volkswagen Foundation (99535 to P.W.), the Federal Ministry of Education and Research (BMBF, to P.W.), and the Boehringer Ingelheim Foundation (Exploration Grant to P.W.). The authors thank all members of the Wu group for discussions and the staff at the MPI Dortmund and TU Dortmund for practical assistance. The authors thank Dr. Sonja Sievers and the Compound Management and Screening Center (COMAS) for assistance in the compound screening, Dr. Fubao Huang for synthesizing compounds KIRA8 and G1749, Ms. Huzhuyue Xie for culturing the Sf9 cells, the Crystallography and Biophysics facility (ZE-CB) team for assisting in protein crystallography and biophysical measurements, and Christiane Heitbrink for measuring the HRMS. The authors thank the European Synchrotron Radiation Facility (ESRF, Grenoble, France) for the beamtime. A.A. acknowledges a doctoral scholarship from DAAD. Y.L., A.A., and L.W. acknowledge the International Max Planck Research School for Living Matter, Dortmund, Germany.

## Author contributions

P.W. conceived and supervised the project. P.W., Y.L., and A.A. designed the experiments. Y.L. performed small-molecule screening, validations, and mechanism studies. P.W., A.A., and Y.L. designed the compounds. A.A. and M.J. synthesized and characterized the small molecules. Y.L. and R.G. performed crystallography studies. Y.L., L.W., and O.H. performed cellular experiments. All authors interpreted the results. The manuscript was written by Y.L., A.A., and P.W. with contributions from all other authors.

## Funding

## Competing interests

The authors declare no competing interests.
