## [Transparent Peer Review file · Nature Communications]

Harnessing Indole Scaffolds to Identify Small-molecule IRE1 α Inhibitors Modulating XBP1 mRNA Splicing

Corresponding Author: Professor Peng Wu

Version 0:

Reviewer comments:

Reviewer #1

(Remarks to the Author)

This manuscript describes the identification and characterization of small molecules that inhibit IRE1 α 's RNase activity through its ATP-binding site. The work represents a substantial medicinal chemistry effort, spanning analog synthesis to structural characterization. The most advanced analog, IA07, demonstrates impressive biochemical potency and selectivity. However, this work does not represent a significant advance for the field. While the authors state that 'the current collection of IRE1 α inhibitors suffered from poor activity and unfavorable selectivity,' highly potent and selective inhibitors of IRE1 α 's RNase activity already exist. For example, KIRA8 (which the authors cite) is highly selective for IRE1 α with few off-targets. Although KIRA8 has poor oral bioavailability, Genentech has developed analogs (see PMID: 38748820) that are highly selective, potent, and orally bioavailable. Moreover, to achieve sufficient potency in cultured cells with their most advanced analog, the authors must use an ethyl ester prodrug—a major liability that likely precludes in vivo use. Therefore, the efforts described here do not significantly advance the development of IRE1 α modulators with more favorable pharmacological properties.

The discovery that IA07 modulates IRE1 α 's ATP-binding site in an unexpected way to achieve RNase inhibition is intriguing and could be of general interest. However, the biochemical and biophysical characterization of IA07's inhibitory mechanism is modest and does not sufficiently illuminate how inhibition occurs. While the co-crystal structure provides a starting point, it alone cannot explain IA07's mechanism or guide the design of alternative inhibitors that exploit similar mechanisms without relying on a problematic propionic acid moiety.

There are also a number of technical issues that need to be addressed prior to publication:

-For Figures 2 and 4, there is a large discrepancy between the biochemical potencies obtained through different methods for IA01 and IA07, respectively. For both compounds, the IC₅₀ obtained in the FRET-based IRE1 α RNase activity assay is approximately 10-fold lower than in the gel-based assay. The potencies of both compounds in the differential scanning fluorimetry (DSF) assay appear to be approximately 50-fold lower than in the FRET-based IRE1 α RNase activity assay (though an exact value cannot be determined since the DSF data were not fit). Both compounds also show unexpectedly high (micromolar) KD values when measured by isothermal titration calorimetry (ITC).

-Using Refeyn mass photometry of cross-linked IRE1 α complexes, the authors observe that IA07 does not reduce IRE1 α dimerization. However, this finding appears inconsistent with their crystallographic data showing separated RNase domains in dimeric IRE1. Since both kinase and RNase domain interactions contribute to the IRE1 dimer interface, one would expect at least some reduction in dimerization levels. It is also a bit worrying that the authors observe that one of the control compounds tested (G1749) also blocks dimerization/oligomerization in their assay (Extended Data Fig. 6d and 6e), as this compound was found not to affect the monomer/dimer equilibrium of IRE1 in PMID: 33318494.

-The authors use a cellular thermal shift assay to demonstrate that the ester analog of IA07 inhibits IRE1 α 's RNase activity (measured as spliced XBP1 levels) in cells through engagement of IRE1a. However, only a single high concentration (20 μ M) of IA07 was tested. Demonstration that the same concentrations of IA07 that prevent thermal denaturation of IRE1a also block its RNase activity would be much more convincing.

Reviewer #2

(Remarks to the Author)

This paper reports on the identification and subsequent characterisation of an indole-based IA07 inhibitor of IRE1. The inhibitor binds to the kinase active site and inhibits the RNase activity of IRE1 located at the C-terminus of the protein in an allosteric fashion. They present in vitro activity and biophysical interaction analysis, followed by co-crystal structures with inhibitor bound, and then cellular analysis of the compound.

The invitro characterisation of the inhibitors using the FRET and other biophysical assay is nice and well conducted with the conclusions well supported by the data. They conclude that the inhibitors work in an allosteric fashion in solution, which is supported by the data.

The major concerns for me are the interpretation of the crystal structures and how the authors deduce these structures to represent an allosteric mechanism of action for this inhibitor and protein. One could argue that the very small shifts in the alignment of dimers within the crystal is due to packing of the crystal lattice and not any allosteric change. They do not mention this alternative interpretation of the data. Also, they make inferences about IRE1 oligomerisation and measure the oligomeric state of cytosolic portion of IRE1 with both IA07 and KIRA8 compound. However, the oligomerisation of IRE1 is driven by the luminal domain in response to ER stress. Since they do not have this domain present in their construct their interpretation of mechanism is not supported by the data. At no stage do they mention this short coming. These short comings impact on their validity of mechanism of action and the impact of the study.

Other points for the authors to consider -- there are 4 molecules in the asymmetric unit. The authors chose to ignore molecules 3 and 4. They should show the interactions between all four molecules in the asymmetric unit. It is important to assess how the other molecules interact. This can be presented in extended figure etc. Also, it would be interesting to align the dimers that formed across the unit cell with the dimers that are arranged within the unit cell and to see if the small shifts are repeated or not.

Reviewer #3

(Remarks to the Author)

In the manuscript titled "Harnessing Indole Scaffolds to Identify Small-molecule IRE1 α Inhibitors Modulating XBP1 mRNA Splicing" by Yang Liu and colleagues, the authors present the discovery and development of a new family of inhibitors based on an indole core and targeting the inositol-requiring enzyme 1 alpha (IRE1 α), a key sensor in the unfolded protein response (UPR). IRE1 α regulates ER stress signaling by mediating XBP1u mRNA splicing, leading to the production of the potent XBP1s transcription factor whose role is to restore the RE homeostasis. IRE1 has been a target of rising interest in various diseases over the past decade and a number of molecules with different mode of actions have been described. Following an experimental screening on an in-house collection of ten thousand molecules, the study introduces a series of indole-based small molecules, identifying IA01 as a promising hit. Structural modifications to the indole scaffold on three key positions optimized its properties and potency, leading in fine to IA107, a highly potent and selective IRE1 α inhibitor demonstrating low nanomolar RNase IC50s against dephosphorylated and phosphorylated IRE1 α , respectively. The compound binds competitively to the ATP-binding pocket of the IRE1 α kinase domain, thereby allosterically inhibiting its RNase activity, but does not affect IRE1 α dimerization in contrast of known inhibitors such as the KIRAs, PAIRs and some of the drugs from Genentech. A co-crystal structure of IA107 in complex with IRE1 obtained at a 3Å resolution allowed the authors to get insights in the mode of action, involving separation of RNase domain interfaces. Finally, in a cellular model, IA107 inhibited ER stress-induced XBP1 mRNA splicing, but was limited by its free carboxylic acid. An ethyl ester prodrug, IAPD1, was prepared and showed improved cellular activity by 50-fold.

Overall, this is a very thorough and well conducted work that should satisfy any medicinal chemists reading the paper and reassure biologists that this molecule represents a well-characterized new tool. The techniques employed and data are of good quality and well presented, and support most of the claims made (more on that below). However, there is a number of points that need to be discussed and/or addressed as they strongly impact the significance of the work for the field, and therefore its publication in this journal and not a more specialized journal like JMedChem, where it would perfectly fit.

My immediate recommendation is therefore major revisions.

Major points:

1. The main argument for novelty is that IA107 and related compounds present a novel mode of action in their inhibition of IRE1 RNase activity (Lys-Glu bridge conserved, dimer not disrupted, RNase domain spread apart) and I am not so sure it is the case. The authors focus their comparison on compounds derived from Amgen study compound 18 (Harrington et al., 2015), also known as KIRA8, and more recent derivatives developed by Genentech, which are molecules known to interfere with the Lys-Glu bridge and disrupt IRE1 dimer (in some cases). However, it would be relevant to compare it to staurosporine instead, for which a crystal structure has been published in 2015 (Concha et al., 2015; PDB ID 4YZC). In this crystal structure, IRE1 exists as a dimer and the alphaC helix is in a similar position as in the authors' structure. Unfortunately, Glu612 is not fully resolved, but its orientation and distance to Lys599 makes it likely that the Lys-Glu bridge is intact. As a matter of fact, when prepared with Schrodinger's preparation wizard and minimized, the bridge is indeed

predicted to exist. In view of this, I would like the authors to do the following:

-compare their structure 9GOW to 4YZC with quantitative indicators (e.g. figure 5E of Concha et al., 2015) and aligned on the kinase domain to compare the position of the RNase domains in both structures.

- perform a new DSS-crosslinking experiment with staurosporine and compare the profile obtained to the one of IA107 and KIRA8.

Together, this should be informative on whether or not the MoA of IA107 is truly original.

2. The second major point that bothers me is the discrepancy between the ~1 micromolar Kds of IA107 (measured by ITC and MST) and the low nanomolar IC50s found the IRE1 RNase assays. With a Kd of around 1 μ M, the fraction of ligand bound in the low nanomolar range is zero or near zero (as seen in figure 4f), which raises the question how can it have an inhibitory effect in this concentration range!? How do the authors explain that? The sentence Lines 333-335 page 17 is not convincing as the correlation between the kinase inhibitory activity and RNase inhibitory activity has been shown to be very strong (see Figure 4 Feldman et al., 2016).

It might be interesting here too to measure the IC50 of the compounds on the kinase activity of IRE1. There are a number of fluorescence, luminescence, or radiometric-based assays available to do that.

3. One last point that require additional experiments is the cell validation. Currently, IA107 efficacy has only been validated in one cell line, A549, whereas its cytotoxicity was evaluated in A549 and three additional cancer cell lines (MDA-MB-468, HCT116 and HT-29). When making such bold statement as "Despite the reported examples, the current collection of IRE1 α inhibitors suffered from poor activity and unfavorable selectivity, which limited the further progression of such inhibitors" (L60-62, p4), the least would be to validate much more in depth your lead compound. Putting down other people's work is never a good look, especially when there is a molecule in phase II clinical trial (ORIN1001/MKC8866) or other very selective and potent IRE1 kinase inhibitors (Braun et al. 2024, JMedChem) Without going as far as requiring ADME and in vivo validation, more testing on other cell lines needs to be done.

Minor points:

Most of the ITC experiments with the IA compounds do not capture the full S curve and could use some optimization of the conditions. Playing with the concentrations of the protein or drugs and doing dual injection runs might be worth pursuing. Additionally, please provide the thermodynamic data of the ITC runs.

Please find below some minor points to the authors. I've also provided an annotated manuscript with comments regarding minor issues to address such as typos, references to add, questions, etc. The scheme of IA84 in the SI is incorrect (it is KIRA8).

Version 1:

Reviewer comments:

Reviewer #2

(Remarks to the Author)

The two major concerns for me were that the authors hadn't considered an alternative interpretation that the small shifts in the structure could be due to crystal packing. They have now added a statement to the manuscript and have tried to justify the structures by analysing all four molecules within the asymmetric unit.

The second concern was the in vitro analysis of IRE1 oligomerisation was conducted using protein that was devoid of the luminal domain which is a driver for oligomerisation. Although cross linking in cells was attempted, they were not greatly successful. The use of the cytosolic domain only has now been added as a discussion point with implications for mechanism.

Reviewer #3

(Remarks to the Author)

I am pleased to see that the authors of the manuscript "Harnessing Indole Scaffolds to Identify Small-molecule IRE1 α Inhibitors Modulating XBP1 mRNA Splicing" carried out all the experiments requested by the reviewers, which shared many similar concerns. I don't have major comments anymore, only two minor points:

1- There is a mistake in the structure of Figure 1, panel b. In the bottom right corner of the kinase binder area, it is not temozolomide but Z4P, a ligand reported by Pelizzari-Raymundo in 2023 in iScience.

2- The thermodynamic data in figure 2f of the titration of IA01 are incoherent (N, H and -T S). Something is not right with that experiment.

Once these two points are addressed, I am in favor of publication.

POINT-BY-POINT RESPONSE

(Notes: the original comments from the Reviewers are reproduced before the detailed “Authors’ response”, all of which have been marked by using blue fonts; All page numbers mentioned in the blue font-highlighted “Authors’ response” in this document indicate the corresponding numbers in the revised manuscript and revised supplementary information unless otherwise specified)

Reviewer #1 (Remarks to the Author):

This manuscript describes the identification and characterization of small molecules that inhibit IRE1 α 's RNase activity through its ATP-binding site. The work represents a substantial medicinal chemistry effort, spanning analog synthesis to structural characterization. The most advanced analog, IA07, demonstrates impressive biochemical potency and selectivity. However, this work does not represent a significant advance for the field. While the authors state that 'the current collection of IRE1 α inhibitors suffered from poor activity and unfavorable selectivity,' highly potent and selective inhibitors of IRE1 α 's RNase activity already exist. For example, KIRA8 (which the authors cite) is highly selective for IRE1 α with few off-targets. Although KIRA8 has poor oral bioavailability, Genentech has developed analogs (see PMID: 38748820) that are highly selective, potent, and orally bioavailable. Moreover, to achieve sufficient potency in cultured cells with their most advanced analog, the authors must use an ethyl ester prodrug—a major liability that likely precludes in vivo use. Therefore, the efforts described here do not significantly advance the development of IRE1 α modulators with more favorable pharmacological properties.

We thank the Reviewer for the general comments regarding our substantial work in evaluating the new IRE1 α inhibitors, as well as the raised concerns. For the statement kindly pointed out by Reviewer that “The current collection of IRE1 α inhibitors suffered from poor activity and unfavorable selectivity...”, we have now revised the description as “Despite the reported IRE1 α inhibitors with impressive potency and encouraging progress, a novel inhibitor chemotype with a distinct inhibitory mechanism may offer a new therapeutic option in addressing related diseases” to more accurately reflect the current landscape of the research in the field. We describe in the following pages that our compounds have a different inhibition mechanism from that of KIRA8. We further described our obtained collective data that would possibly convince the readers that this work would be a significant advance contributing to the field of IRE1 inhibitor and modulator development.

The discovery that IA07 modulates IRE1 α 's ATP-binding site in an unexpected way to achieve RNase inhibition is intriguing and could be of general interest. However, the biochemical and biophysical characterization of IA07's inhibitory mechanism is modest and does not sufficiently illuminate how inhibition occurs. While the co-crystal structure provides a starting point, it alone cannot explain IA07's mechanism or guide the design of alternative inhibitors that exploit similar mechanisms without relying on a problematic propionic acid moiety.

We appreciate that the Reviewer commented on the intriguing inhibitory mechanism of IA107. The allosteric inhibition mechanism of IA107 was concluded based on the combined evidence from the resolved co-crystal structure and data from a suite of biophysical measurements and biochemical assays. IA107 binds to the kinase site, which is distinct from the RNase active site, but inhibits the RNase activity of IRE1 α . The enzyme kinetic study indicated a non-competitive inhibition mode towards an *XBPI* RNA substrate. Together with the structural changes induced by the compound illuminates the allosteric inhibition mode of IA107. Furthermore, the effect of IRE1 α -regulated downstream and target engagement were evaluated in cellular assays.

There are also a number of technical issues that need to be addressed prior to publication:
-For Figures 2 and 4, there is a large discrepancy between the biochemical potencies obtained through different methods for IA01 and IA07, respectively. For both compounds, the IC50 obtained in the FRET-based IRE1 α RNase activity assay is approximately 10-fold lower than in the gel-based assay. The potencies of both compounds in the differential scanning fluorimetry (DSF) assay appear to be approximately 50-fold lower than in the FRET-based IRE1 α RNase activity assay (though an exact value cannot be determined since the DSF data were not fit). Both compounds also show unexpectedly high (micromolar) KD values when measured by isothermal titration calorimetry (ITC).

We thank the Reviewer for the remarks regarding the technical issues. The activities obtained in the FRET assay and gel-based assay were different due to varied assay sensitivity and assay conditions. The FRET assay measured the inhibition kinetically over 30 min, and the linear part of the curve was adopted and calculated, but the gel cleavage assay only measured the reaction endpoint after 45 min incubation. In the FRET-based assay, we used a final concentration of 4 nM p-IRE1 α protein and a final concentration of 200 nM FAM-XBP1-BHQ RNA, and for gel-based cleavage assay, to have a visible band and reasonable resolution, the final concentrations of 10 nM p-IRE1 α protein and 50 nM FAM-XBP1-BHQ RNA were used. The FRET assay is more sensitive in detecting cleavage in comparison with the gel-based assay, as a minor amount of RNA cleavage will lead to the FRET signal change. Whereas in the gel-based cleavage assay, a relatively larger amount of RNA needs to be cleaved to yield a visible cleaved RNA band on the gel. For better comparison, the gel-based cleavage assay using a final concentration of 4 nM p-IRE1 α protein and a final concentration of 200 nM FAM-XBP1-BHQ RNA was performed (Fig. R1A), which showed weaker cleaved RNA bands but gave an activity closer to the FRET-assay using the same protein and RNA concentrations.

The DSF assay measures the thermostability changes upon ligand binding, it is an indirect measurement, and the correlation of T_m shift with the enzymatic-based activity is weak (PMID24915177). In this manuscript, we used the Boltzmann model to identify the melting temperature (T_m). However, it did not provide more thermodynamic transition information for K_D determination. We didn't try to fit the DSF data to calculate the K_D using more complicated analysis methods because K_D is defined under isothermal conditions, and DSF assay is measured under a temperature gradient. In our manuscript, the DSF assay was used as a qualitative method to determine the binding between protein and ligand.

The explanation for the discrepancy between K_D and IC_{50} of IA compounds might be that the compounds prefer to bind to the dimerized proteins to yield the inhibitory activity, and only the dimeric form of IRE1 α harbors the active RNase site and exerts the RNase activity. To test this hypothesis, the binding affinity of IA107 against phosphorylated IRE1 α (which contains a higher ratio of dimer + oligomeric protein vs monomer than that of the dephosphorylated IRE1 α) was evaluated by MST assay using the fluorophore-labeled protein. In this assay, IA107 indeed showed a higher binding affinity against the p-IRE1 α with a K_D value of 152 nM (Fig. R1B), the data was added to the revised manuscript Fig.4g. To address this point, we added the following discussion to the current manuscript on page 9:

“The binding affinity of compound IA107 against p-IRE1 α was measured by MST, which showed an increased affinity with a K_D of 152 nM. The different binding affinity (K_D) against different IRE1 α states could be explained by IA107 preferring to bind to the dimeric form of proteins, which yields the RNase inhibitory activity. The p-IRE1 α assembles more dimers and therefore has a higher affinity to IA107 in comparison with that of IRE1 α . This may also explain the observed discrepancy between the inhibitory IC_{50} and the binding K_D of IA107. Once the dimeric proteins are bound and inhibited, the RNase activity of IRE1 α will be inhibited since the monomeric proteins are inactive.”

And the following discussion has now been added to the “Conclusion” section of the revised manuscript on page 20 to elaborate on the possible reasons contributing to such discrepancy:

“IA107 may preferentially bind to the active dimeric proteins, which could cause the different affinities against p-IRE1 α and IRE1 α and the observed discrepancy between the binding affinity and functional inhibitory potency. The discrepancy may also be attributed to the allosteric mechanism of IA107, which binds to the kinase domain and induces conformational changes to inhibit the RNase activity rather than directly competing with the substrate. After the

dissociation of IA107, IRE1 α may remain in the inactive conformation, contributing to the sustained inhibition effect.”

Fig. R1. (A) Gel-based cleavage assay using the same condition as FRET assay. (B) Fig.4g of current manuscript, IA107 binding affinity towards p-IRE1 α in MST assay.

-Using Refeyn mass photometry of cross-linked IRE1 α complexes, the authors observe that IA07 does not reduce IRE1 α dimerization. However, this finding appears inconsistent with their crystallographic data showing separated RNase domains in dimeric IRE1. Since both kinase and RNase domain interactions contribute to the IRE1 dimer interface, one would expect at least some reduction in dimerization levels. It is also a bit worrying that the authors observe that one of the control compounds tested (G1749) also blocks dimerization/oligomerization in their assay (Extended Data Fig. 6d and 6e), as this compound was found not to affect the monomer/dimer equilibrium of IRE1 in PMID: 33318494.

We appreciate the Reviewer’s comments regarding the dimerization experiments. Our crystal structure shows a conformation with separated RNase domains, while the kinase domains remain in the same dimer interaction interface as the apo p-IRE1 α dimer. However, we did not observe dimerization inhibition by IA107 in the mass photometry assay nor SDS-PAGE gel-based crosslinking assay (Fig. R2A). In contrast, KIRA8 inhibited dimerization in both assays. We think this can be explained by that the kinase domains are more important for the dimer formation. Phosphorylation of the kinase domain greatly enhances the dimerization and forms the RNase active site (PMID: 18191223, PMID: 19079236, PMID: 33318494, PMID: 34911951). By analyzing the interface, we found that the majority of the hydrogen bonds were

formed in the kinase domain. Although the RNase domain active site was more separated and the interaction between D847 and H909 was disrupted, other interactions were formed by structural shift, e.g., the interaction between D847 and R912 (Fig. R2B). The overall interactions between the two dimers were maintained and the dimerization levels were not impacted.

We thank the Reviewer for mentioning the work from Ferri *et al* (PMID: 33318494). Our data and observations are actually consistent with the data included in the work from Ferri *et al*. Extended Data Fig. 6d from our work showed that KIRA8 (the AMG-18 in PMID:33318494) has an IC₅₀ of 0.021 μM against dephosphorylated IRE1α (the “IRE1-0p” in PMID:33318494), and an IC₅₀ of 0.003 μM against p-IRE1α (the “IRE1-3p” in PMID:33318494). This data can be referred to Figure 5a from the reference PMID 33318494 (though the authors did not give clear IC₅₀ values) by looking at the figure and tracking back to their source data, which are very consistent with our data. Extended Data Fig 6e from our manuscript shows that G1749 blocks the dimerization of p-IRE1α in the mass photometry assay, as the Reviewer kindly pointed out. A similar result can be found in Figure 5b of their publication using a different assay, the SV-AUC assay. For an easier understanding, we quote the sentence from their work here, “SV-AUC experiments with AMG-18 and G-1749 in complex with IRE1-3P showed that both compounds inhibit dimer formation (Fig. 5b), suggesting that the tail of G-1749 likely occupies the back pocket of IRE1-3P, thereby disrupting the C-helix position and leading to RNase inhibition. Of note, all compounds derived from G-1749 (Fig. 3) also inhibited IRE1-3P (Supplementary Fig. 5), albeit to different degrees.”. Though the methods used between the report study and this study were different, the same result was observed that G-1749 inhibited the dimerization of phosphorylated IRE1α KR domain protein.

The aforementioned figure and Fig.5b reported by Ferri *et al* (PMID: 33318494) are shown below:

B

Hydrogen bonds found in dimer interface

IA107-p-IRE1 α (9gaw)				Apo-p-IRE1 α (6w3c)			
#	Chain B	Dist. [Å]	Chain A	#	Chain D	Dist. [Å]	Chain B
1	ARG 617 [NH1]	2.87	ASP 592 [OD1]	1	ARG 617 [NE]	2.86	ASP 592 [O]
2	TYR 628 [OH]	2.54	ASP 592 [OD2]	2	THR 631 [N]	3.51	ASP 592 [OD1]
3	THR 631 [N]	3.47	ASP 592 [OD2]	3	ARG 617 [NE]	3.82	ASP 592 [OD2]
4	ARG 627 [NH2]	2.93	ASP 620 [OD2]	4	TYR 628 [OH]	2.68	ASP 592 [OD2]
5	ARG 594 [NH2]	2.41	ASP 620 [OD2]	5	ARG 617 [NH2]	3.60	ASN 593 [OD1]
6	ARG 594 [NH1]	3.34	TYR 628 [O]	6	ARG 627 [NH1]	2.97	ASP 620 [OD1]
7	LYS 568 [NZ]	2.59	THR 631 [O]	7	ARG 627 [NH2]	2.50	ASP 620 [OD2]
8	HIS 702 [NE2]	3.08	SER 681 [O]	8	ARG 627 [NH2]	3.22	TYR 628 [O]
9	ARG 955 [NH1]	2.97	GLU 836 [OE2]	9	LYS 568 [NZ]	3.50	THR 631 [O]
10	ARG 955 [NH2]	2.65	ASP 927 [OD2]	10	LYS 568 [NZ]	3.54	GLU 632 [OE1]
11	ASP 592 [OD1]	3.49	ARG 617 [NE]	11	ARG 955 [NH2]	3.17	GLU 836 [OE1]
12	ASP 592 [OD2]	2.56	TYR 628 [OH]	12	HIS 902 [NE2]	3.45	ASP 847 [OD1]
13	ASP 592 [OD2]	3.40	THR 631 [N]	13	ASP 592 [O]	2.75	ARG 617 [NH1]
14	GLN 614 [OE1]	3.38	ASN 593 [ND2]	14	ASP 592 [OD1]	3.22	THR 631 [N]
15	ASP 620 [OD2]	2.48	ARG 594 [NH2]	15	ASP 592 [OD2]	2.54	TYR 628 [OH]
16	ASP 620 [OD2]	3.67	ARG 627 [NH2]	16	ASP 592 [OD2]	2.70	ARG 617 [NH1]
17	GLU 621 [OE2]	3.66	ARG 627 [NH1]	17	ARG 617 [O]	3.76	ARG 594 [NH2]
18	TYR 628 [O]	3.71	ARG 594 [NH1]	18	ASP 620 [OD1]	3.22	ARG 594 [NH2]
19	TYR 628 [O]	3.33	ARG 594 [NH2]	19	ASP 620 [OD1]	3.21	ARG 627 [NH2]
20	TYR 628 [O]	3.51	ARG 627 [NH2]	20	ASP 620 [OE2]	3.51	ARG 627 [NH1]
21	PHE 629 [O]	3.72	ARG 594 [NH1]	21	GLU 621 [OE1]	2.88	ARG 627 [NH2]
22	SER 681 [O]	3.15	HIS 702 [NE2]	22	GLU 621 [OE2]	3.35	LYS 706 [NZ]
23	GLU 836 [OE1]	3.34	ARG 955 [NH2]	23	TYR 628 [O]	3.57	ARG 594 [NH1]
24	ASP 847 [O]	3.11	ARG 912 [NH2]	24	THR 631 [O]	2.98	LYS 568 [NZ]
25	GLU 913 [OE1]	3.03	LYS 851 [NZ]	25	GLU 632 [OE1]	3.56	LYS 568 [NZ]
26	ASP 927 [OD1]	3.13	ARG 955 [NH2]	26	GLU 836 [OE1]	2.87	ARG 955 [NE2]
27	ASP 927 [OD2]	3.27	ARG 955 [NH1]	27	ASP 847 [OD1]	3.72	HIS 909 [NE2]
				28	GLU 954 [OE1]	2.59	ARG 912 [NH2]

[editorial note: panel redacted]

Fig. R2. (A) SDS-PAGE gel-based crosslinking assay using different concentrations of p-IRE1 α protein. (B) Hydrogen bonds in the interface of IA107-bound p-IRE1 α dimer and apo p-IRE1 α dimer. Analyzed using the PDBePISA web server (PMID: 17681537). The interactions mentioned in the response are highlighted. (C) Fig. 5b of Ferri *et al.*

-The authors use a cellular thermal shift assay to demonstrate that the ester analog of IA07 inhibits IRE1 α 's RNase activity (measured as spliced XBP1 levels) in cells through engagement of IRE1 α . However, only a single high concentration (20 μ M) of IA07 was tested. Demonstration that the same concentrations of IA07 that prevent thermal denaturation of IRE1 α also block its RNase activity would be much more convincing.

We thank the Reviewer for the useful suggestion of performing a dose-dependent cellular thermal shift assay. The assay has now been performed with more concentrations of IAPD1 and the result has been added to the revised manuscript as Fig.7e. Additionally, we evaluated IA107 using A549 cell lysate, the result of which has now been added to the revised manuscript as the Extended Data Fig. 7a. The result showed that IA107 stabilized full-length IRE1 α in cell lysate dose-dependently, and IAPD1 dose-dependently stabilized the IRE1 α in cells. We found that the stabilizing effects of IAPD1 are negligible when the treatment concentrations were lower than 2.22 μ M. As the thermal stability changes upon ligand binding are indirect measurements, the correlation with the enzymatic-based activity is weak (PMID24915177). Hence, the concentrations for the thermal shift and activity assays are different. To elaborate on this point, the following description has now been added to the revised manuscript on page 17:

“The binding of IA107 to the wild-type full-length IRE1 α was evaluated by a thermal stability assay using A549 cell lysate in western blot analysis (Extended Data Fig. 7a). The result showed that IA107 stabilized the full-length IRE1 α in a concentration-dependent manner, indicating the binding of IA107 to the wild-type IRE1 α .”

Page 18: “...which demonstrated that the thermal stability of cellular IRE1 α was concentration-dependently increased in comparison with that of the DMSO control upon IAPD1 treatment...”

The aforementioned figures stand as:

Extended Data Fig 7a: IA107 concentration

-dependently stabilized full-length IRE1 α protein in A549 cell lysate. **Fig. 7e:** Cellular thermal shift assay showing IAPD1 dose-dependently stabilized IRE1 α in A549 cells upon the treatment with different concentrations of IAPD1 or DMSO for 4 h.

Reviewer #2 (Remarks to the Author):

This paper reports on the identification and subsequent characterisation of an indole-based IA07 inhibitor of IRE1. The inhibitor binds to the kinase active site and inhibits the RNase activity of IRE1 located at the C-terminus of the protein in an allosteric fashion. They present in vitro activity and biophysical interaction analysis, followed by co-crystal structures with inhibitor bound, and then cellular analysis of the compound.

The invitro characterisation of the inhibitors using the FRET and other biophysical assay is nice and well conducted with the conclusions well supported by the data. They conclude that the inhibitors work in an allosteric fashion in solution, which is supported by the data.

We appreciate the Reviewer for the positive comments and thoughtful suggestions regarding our manuscript. Please see below for our detailed responses to the Reviewer's kindly raised specific questions.

The major concerns for me are the interpretation of the crystal structures and how the authors deduce these structures to represent an allosteric mechanism of action for this inhibitor and protein. One could argue that the very small shifts in the alignment of dimers within the crystal is due to packing of the crystal lattice and not any allosteric change. They do not mention this alternative interpretation of the data. Also, they make inferences about IRE1 oligomerisation and measure the oligomeric state of cytosolic portion of IRE1 with both IA07 and KIRA8 compound. However, the oligomerisation of IRE1 is driven by the luminal domain in response to ER stress. Since they do not have this domain present in their construct their interpretation of mechanism is not supported by the data. At no stage do they mention this short coming. These short comings impact on their validity of mechanism of action and the impact of the study.

We thank the Reviewer for the insightful comments. We agree with the Reviewer that we cannot fully exclude that crystal packing may contribute to the observed structure shift. However, the allosteric regulation mechanism of IA107 was concluded not only based on the structural conformational changes. Firstly, IA107 binds to the kinase site, which is distinct from the RNase active site, but inhibits the RNase activity. Secondly, IA107 showed a non-competitive

inhibition mode towards the *XBPI* RNA substrate in the enzyme kinetic assay (Fig. 5a) (PMID: 31424826). Both aspects indicated an allosteric inhibition mechanism. Together with the observed ligand-induced conformational changes in the resolved structure, the IA compounds were thus classified as allosteric inhibitors.

We acknowledge the Reviewer's concern regarding the crystal packing effects. To discuss this alternative interpretation, as well as to outline the allosteric mechanism of IA107 that is concluded based on the combined results, including structural shifts and biochemical and biophysical data, we have now revised our manuscript in the structure section on page 15 by adding the following description:

“Overall, the resolved complex structure between p-IRE1 α and IA107 further supports the allosteric inhibition mechanism of IA107, in line with the non-competitive inhibition mode against the *XBPI* RNA substrate. While crystal packing may cause small shifts of the side chains, the structure of IA107-bound p-IRE1 α complex suggests the ligand-induced conformational change upon binding to the IRE1 α kinase site, which matched the allosteric inhibition mode suggested by the biochemical and biophysical data.”

We agree with the Reviewer that the luminal domain is important for IRE1 α oligomerization. As the Reviewer stated, the oligomerization of IRE1 α is driven by the luminal domain in response to ER stress in cells. The chaperone Bip will dissociate from IRE1 α 's luminal domain upon detection of ER stress, which triggers the dimerization and autophosphorylation of the cytosolic domains. Therefore, many studies on IRE1 α modulator and IRE1 α biology have used the cytosolic domains to evaluate the protein dimerization in in vitro assays (PMID 27227314, PMID 25968568, PMID 33318494, PMID 34911951, PMID: 23086298). Additionally, we used the cytosolic portion to understand the impact of IA107 against the dimerization of IRE1 α due to the practical constraints on purifying the full-length protein. To further investigate the dimerization effect of IA compounds involving the luminal domain of IRE1 α , we performed the crosslinking assay in cell lysate after compound treatments, by following an adjusted protocol based on PMID33318494 (Fig. R3). However, the dimer/oligomer bands of IRE1 α in the western blot could not be observed even for the tunicamycin-treated groups, which might be due to the commercialized IRE1 α antibody lacking affinity to the dimers. Despite that, we observed that IA107 effectively inhibited the expression of XBP1s protein induced by tunicamycin, indicating that IA107 inhibited the IRE1 α -mediated UPR pathway. The western blot analyses without the DSS crosslinking step have been added to the revised manuscript as

Fig. 7d and Extended Data Fig.7i. Correspondingly, the following description has now been added on page 18:

“The translated XBP1s protein and IRE1 α protein levels were evaluated via western blot in both A549 and MDA-MB-231 cells (Fig. 7d, Extended Data Fig.7i). KIRA8 and G1749 were used as reference compounds⁴². Under tunicamycin stimulation, the XBP1s protein concentration increased in both cell lines. The pre-treatment of IA107 and KIRA8 inhibited the ER stress-induced XBP1s protein increment. Meanwhile, treatment of G1749 at either 1 μ M or 5 μ M partially inhibited the XBP1s protein induced by ER stress. The treatment of compounds IAPD1, KIRA8, and G1749 did not affect the IRE1 α protein level, indicating that the observed activity of the compounds against IRE1 α downstream proteins was not due to the direct effect on the IRE1 α protein as degraders.”

On the other hand, we recognize the limitation of using only the cytosolic domain. To discuss this limitation, the following discussion has now been added on page 20 in the conclusion section of the revised manuscript:

“Given that the IA compound did not inhibit the dimerization and oligomerization of the IRE1 α cytoplasmic domain in the *in vitro* assay, we expect that the IA compound would not inhibit the assembly in the cellular context either, since the luminal domain drives the IRE1 α dimerization in cells. Therefore, the impact of IA compounds on IRE1 α assembly in a cellular context warrants further investigation to fully understand the inhibitory mechanism.”

The aforementioned Figures stand as:

Fig. R3. (A) Cell lysate crosslinking analyses in A549 cells. (B) Cell lysate crosslinking analyses in MDA-MB-231 cells. The cells were treated with indicated compound for 2h, then a final concentration of 1µg/mL tunicamycin was added and incubated for 4h to induce the ER stress. After cells were lysate, the DSS crosslinking was performed and the crosslinked samples were analyzed by western blot.

Fig. 7d: IAPD1 inhibited the XBP1s protein level in A549 cells by western blot assay. Cells were treated with indicated compounds for 2 h, then a final concentration of 1µg/mL tunicamycin was added into the cells and incubated for another 4 h before cells were lysed. **Extended Data Fig. 7i:** IAPD1 inhibited the XBP1s protein level in MDA-MB-231 cells by western blot assay.

Other points for the authors to consider -- there are 4 molecules in the asymmetric unit. The authors chose to ignore molecules 3 and 4. They should show the interactions between all four molecules in the asymmetric unit. It is important to assess how the other molecules interact. This can be presented in extended figure etc. Also, it would be interesting to align the dimers that formed across the unit cell with the dimers that are arranged within the unit cell and to see if the small shifts are repeated or not.

We thank the Reviewer for the suggestion. The ligand interactions, the K599-E612 salt bridges, DFG motifs, phosphorylated activation loops, and outward pushed N-terminal α C-helices of all chains were presented in Supplementary Fig. 1b,1c,1d, and 1e, respectively. All four molecules showed similar conformational changes against the evaluated structural elements. Ligand IA107 formed hydrogen bonds with C645, K599, and D711 in all chains, the DFG motifs showed DFG-in conformation, and the N-terminal α C-helices were slightly pushed out, with the K599-E612 salt bridges remaining intact. We added the following description in the main text of the revised manuscript on page 14:

“The intact K599 and E612 salt bridge, DFG-in conformation, phosphorylated activation loop and outward pushed N-terminal α C-helix of chains A, C and D were present in Supplementary Fig. 1b-e. The four copies in the asymmetric unit showed similar conformational changes for the evaluated structure elements upon IA107 binding.”

In the structure, chains A, B, and C showed good density. The RNase domain of chain D was not well resolved due to the poor electron density, so we decided to exclude residues K851 to M948 from our model. For the same reason, the dimer formed by chain C with the symmetry mate chain D has a considerably higher B factor compared to the dimer formed between chain A and chain B (Fig. R4A), therefore making the analyses of AB dimer more reliable. We analyzed the kinase domain interaction residues of the CD dimer interface (Fig. R4B), which showed only minimal changes of the key interface residues in comparison with those of the AB dimer. For this reason, we only analyzed the AB dimer in the originally submitted manuscript. The chain D deletion information was previously provided in the method section, which we have now added in the main text in the structure section on page 14 of the revised manuscript as

“In the complex structure, chains A, B, and C showed good density. As the RNase domain of chain D was not well resolved and the residues K851 and M948 were eliminated due to the poor electron density, we mainly focused on the dimer formed by chain A and chain B in the following analysis and discussion.”

The aforementioned Supplementary Fig. 1 of the revised manuscript stands as:

Supplementary Fig. 1. Interaction and conformation of other chains in the asymmetric unit of the resolved IA107-p-IRE1 α complex structure. (a) Ligand interactions between chains A, C, and D. (b) Superimposition view of the α C-helix and K599-E612 salt bridge of all chains. (c) Superimposition view of the DFG motif of all chains. (d) Superimposition view of the activation loop and phosphorylation sites of all chains. (e) Superimposition view of the α C-helix from chains A, C, and D of the IA107-p-IRE1 α complex structure with the apo-pIRE1 α (grey, PDB 6w3c). Chain A is shown in orange, chain B in wheat, chain C in cyan, and chain D in aquamarine.

Fig. R4. (A) The B factors of the AB dimer (left) and CD dimer (right). (B) Superimposition view of the kinase domain interface residues of the AB dimer and CD dimer. Chain A is shown in orange, chain B in wheat, chain C in cyan, and chain D in aquamarine.

Reviewer #3 (Remarks to the Author):

In the manuscript titled "Harnessing Indole Scaffolds to Identify Small-molecule IRE1 α Inhibitors Modulating XBP1 mRNA Splicing" by Yang Liu and colleagues, the authors present the discovery and development of a new family of inhibitors based on an indole core and targeting the inositol-requiring enzyme 1 alpha (IRE1 α), a key sensor in the unfolded protein response (UPR). IRE1 α regulates ER stress signaling by mediating XBP1u mRNA splicing, leading to the production of the potent XBP1s transcription factor whose role is to restore the RE homeostasis. IRE1 has been a target of rising interest in various diseases over the past decade and a number of molecules with different mode of actions have been described.

Following an experimental screening on an in-house collection of ten thousand molecules, the study introduces a series of indole-based small molecules, identifying IA01 as a promising hit. Structural modifications to the indole scaffold on three key positions optimized its properties and potency, leading in fine to IA107, a highly potent and selective IRE1 α inhibitor demonstrating low nanomolar RNase IC50s against dephosphorylated and phosphorylated IRE1 α , respectively. The compound binds competitively to the ATP-binding pocket of the IRE1 α kinase domain, thereby allosterically inhibiting its RNase activity, but does not affect IRE1 α dimerization in contrast of known inhibitors such as the KIRAs, PAIRs and some of the drugs from Genentech. A co-crystal structure of IA107 in complex with IRE1 obtained at a 3Å resolution allowed the authors to get insights in the mode of action, involving separation of RNase domain interfaces. Finally, in a cellular model, IA107 inhibited ER stress-induced XBP1 mRNA splicing, but was limited by its free carboxylic acid. An ethyl ester prodrug, IAPD1, was prepared and showed improved cellular activity by 50-fold.

Overall, this is a very thorough and well conducted work that should satisfy any medicinal chemists reading the paper and reassure biologists that this molecule represents a well-characterized new tool. The techniques employed and data are of good quality and well presented, and support most of the claims made (more on that below). However, there is a number of points that need to be discussed and/or addressed as they strongly impact the significance of the work for the field, and therefore its publication in this journal and not a more specialized journal like JMedChem, where it would perfectly fit.

We genuinely appreciate the Reviewer for the generous comments and constructive suggestions regarding our manuscript. Please see our detailed response to the Reviewer's specific questions below.

My immediate recommendation is therefore major revisions.

Major points:

1. The main argument for novelty is that IA107 and related compounds present a novel mode of action in their inhibition of IRE1 RNase activity (Lys-Glu bridge conserved, dimer not disrupted, RNase domain spread apart) and I am not so sure it is the case. The authors focus their comparison on compounds derived from Amgen study compound 18 (Harrington et al., 2015), also known as KIRA8, and more recent derivatives developed by Genentech, which are molecules known to interfere with the Lys-Glu bridge and disrupt IRE1 dimer (in some cases). However, it would be relevant to compare it to staurosporine instead, for which a crystal structure has been published in 2015 (Concha et al., 2015; PDB ID 4YZC). In this crystal structure, IRE1 exists as a dimer and the alphaC helix is in a similar position as in the authors' structure. Unfortunately, Glu612 is not fully resolved, but its orientation and distance to Lys599 makes it likely that the Lys-Glu bridge is intact. As a matter of fact, when prepared with Schrodinger's preparation wizard and minimized, the bridge is indeed predicted to exist. In view of this, I would like the authors to do the following:

-compare their structure 9GOW to 4YZC with quantitative indicators (e.g. figure 5E of Concha et al., 2015) and aligned on the kinase domain to compare the position of the RNase domains in both structures.

- perform a new DSS-crosslinking experiment with staurosporine and compare the profile obtained to the one of IA107 and KIRA8.

Together, this should be informative on whether or not the MoA of IA107 is truly original.

We thank the Reviewer for the suggestion of comparing the structure 9GOW to 4YZC and performing a DSS-crosslinking experiment with staurosporine (STS). We agree that the Glu612 of 4YZC is not fully resolved, while the Lys-Glu bridge is intact upon STS binding as shown in Fig. R5A. However, the modulation property of STS against IRE1 α RNase activity are very different from that of compound IA107. STS did not affect the RNase activity of phosphorylated IRE1 α but activated the RNase activity of dephosphorylated IRE1 α , which was reported by Concha et al. (Fig.4A, 4B of Concha et al.) and has been validated by in-house data (Fig. R5B). We quote the following statement from Concha et al.: "We found that staurosporine inhibited pIRE1 α kinase enzymatic activity (IC₅₀=3 nM) but had no effect on pIRE1 α RNase activity in vitro (Fig. 4A). STS exhibited a similar binding affinity toward phosphorylated IRE1 α (IC₅₀=30 nM) and dephosphorylated IRE1 α (IC₅₀=50 nM) in a competitive binding

assay (data not shown). Interestingly, incubation of STS with dephosphorylated (inactive) IRE1 α was capable of activating IRE1 α RNase activity (Fig. 4B).”

IA107 inhibited both phosphorylated IRE1 α and dephosphorylated IRE1 α . Similar to the binding of IA107, the kinase domain interface upon STS binding is very similar to the apo form (Fig. R5C), and the p-IRE1 α RNase domain upon STS binding is also identical to that of the apo-p-IRE1 (Fig. R5D), which could explain that STS does not affect p-IRE1 α RNase activity. The interaction interface areas were calculated using the PDBePISA web server (E. Krissinel and K. Henrick, 2007). The quantitative indicators are listed in Figure R5E. The result shows that in comparison with the apo form, the interface area of the kinase domain upon IA107 binding remained almost the same with a 1.63% decrease compared to the apo form. In contrast, the RNase domain interface area upon IA107 binding led to a 26.23% decrease. To our surprise, all interface areas decreased dramatically upon STS binding in comparison with that of the apo form. However, we found that some side chains in the STS-p-IRE1 α structure were not modelled and hence missing in the interface between the RNase domain (Fig. R5F) and the kinase domain (Fig. R5G), which reduces the calculated interface area and makes it less comparable to the structure obtained in this study or to the reported apo form structure. The quantitative interface area for IA107-bound p-IRE1 α and apo form p-IRE1 α were added to the revised manuscript as Extended Data Fig. 6d. Additionally, the following description has been added to the main text on page 14:

“The interaction interface areas of the dimers were analyzed using the PDBePISA web server, which showed only a minimal change up to 1.63% decrease in the kinase domain interface in comparison with the apo form dimer. For the RNase domain of dimer AB, the interaction interface decreased by ~26.2% (Extended Data Fig. 6d)”

A DSS-crosslinking assay was performed with STS (Fig. R5H), which showed that both STS and IA107 did not affect the dimerization and oligomerization of p-IRE1 α . However, STS did not inhibit the RNase activity of p-IRE1 α but IA107 did.

In conclusion and to summarize the above-described points. For the kinase domain part, the p-IRE1 α conformational changes upon STS binding and IA107 binding were similar in the kinase domain, and the dimerization and oligomerization properties of p-IRE1 α were not affected by either STS or IA107. For the RNase domain part, whereas STS did not impact the key RNase domain dimer interaction of p-IRE1 α and had no impact on the RNase activity of p-IRE1 α , IA107 led to misplaced key residues and inhibited the RNase activity of p-IRE1 α .

The aforementioned figures stand as:

Fig. R5. (A) Superimposition of the α C-helix from the IA107-bound p-IRE1 α (wheat, PDB 9gow), apo-pIRE1 α

(grey, PDB 6w3c), and STS-pIRE1 α (green, PDB 4yzc). (B) STS activities against IRE1 α tested in-house. (C) Superimposition view of the kinase dimer interaction from the STS-p-IRE1 α (green, PDB 4yzc) and apo-pIRE1 α (grey, PDB 6w3c). (D) Superimposition view of the RNase dimer interaction from the STS-p-IRE1 α (green, PDB 4yzc) and apo-pIRE1 α (grey, PDB 6w3c). (E) Quantitative interface areas were calculated using the PDBePISA web server. (F) Missing side chains in the STS-p-IRE1 α structure RNase domain interface. STS-pIRE1 α (4yzc) is shown in green, apo-pIRE1 α (6w3c) is shown in grey. (G) Missing side chains in the STS-p-IRE1 α structure kinase domain interface. (H) DSS-crosslinking assay with compound staurosporine, IA107, and KIRA8.

2. The second major point that bothers me is the discrepancy between the ~1 micromolar Kds of IA107 (measured by ITC and MST) and the low nanomolar IC₅₀s found in the IRE1 RNase assays. With a Kd of around 1 μ M, the fraction of ligand bound in the low nanomolar range is zero or near zero (as seen in figure 4f), which raises the question of how can it have an inhibitory effect in this concentration range!? How do the authors explain that? The sentence Lines 333-335 page 17 is not convincing as the correlation between the kinase inhibitory activity and RNase inhibitory activity has been shown to be very strong (see Figure 4 Feldman et al., 2016).

It might be interesting here too to measure the IC₅₀ of the compounds on the kinase activity of IRE1. There are a number of fluorescence, luminescence, or radiometric-based assays available to do that.

We thank the Reviewer for the comment and the suggestion of measuring the kinase activity of the inhibitors. As suggested by the Reviewer, we further tested the IC₅₀ of the IA compounds against the kinase activity with the LanthaScreen™ Eu kinase binding assay by SelectScreen Kinase Profiling Services (Thermo Fisher). The results were added to the revised manuscript Fig. 5e and the following description has been added to the revised main manuscript (page 11) as:

“The kinase inhibitory activities of selected IA compounds (IA01, IA06, IA10, IA64, and IA107) against IRE α were measured using LanthaScreen™ Eu kinase binding assay (Fig. 5e). Compound IA107 (RNase activity: dephosphorylated IRE1 α , IC₅₀: 16 nM; p-IRE1 α , IC₅₀: 9 nM) showed the most potent inhibition among the tested IA compounds with an IC₅₀ of 768 nM (kinase activity). Compound IA64 (RNase activity: dephosphorylated IRE1 α , IC₅₀: 30 nM; p-IRE1 α , IC₅₀: 30 nM) showed only slightly decreased kinase activity (IC₅₀: 1104 nM) in comparison with that of IA107. Compounds IA01 (RNase activity: dephosphorylated IRE1 α , IC₅₀: 300 nM; p-IRE1 α , IC₅₀: 320 nM) and IA10 (RNase activity: dephosphorylated IRE1 α , IC₅₀: 1810 nM; p-IRE1 α , IC₅₀: 1970 nM) showed weaker kinase inhibitory potency with IC₅₀ of 2470 nM and 9610 nM, respectively. Compound IA06 (RNase activity: dephosphorylated IRE1 α , IC₅₀: 7.32 nM; p-IRE1 α , IC₅₀: 7.79 nM) showed the least active kinase inhibitory

activity, with only 23% inhibition at the highest tested concentration of 50 μ M. In a word, the kinase inhibition results further confirmed the binding of IA107 to the kinase pocket of IRE1 α , together with the observation of a positive correlation between the kinase and RNase activities of the IA compounds”.

For the IA compounds in this study, we did not observe the linear correlation between the kinase activity and RNase activity as reported for the KIRAs compounds by Feldman et al., 2016. Instead, we observed a non-linear but positive correlation. The difference could be attributed to the different mechanisms of inhibition of the IA compounds in comparison with the reported KIRA compounds. The KIRA compounds inhibited the dimerization of IRE1 α , whereas the IA compounds did not. Notably, the kinase and RNase activities of IRE1 α can be regulated independently, e.g. staurosporine has been reported to inhibit the kinase activity of IRE1 α but showed no impact on the RNase activity of phosphorylated IRE1 α (Concha et al., 2015).

The possible reason for the discrepancy between the K_D and IC_{50} of IA compounds might be due to the fact that IA107 prefers to bind to the dimerized IRE1 α . To validate this hypothesis, we tested the binding affinity of IA107 against p-IRE1 α (which contains more dimer/oligomer) by MST assay using a fluorophore-labeled protein, which indeed demonstrated that IA107 showed a higher binding affinity against p-IRE1 α with a K_D value of 152 nM (this data was added to the revised manuscript as Fig. 4g). Correspondingly, we have now added the following description and discussion to the revised manuscript on page X:

“The binding affinity of compound IA107 against p-IRE1 α was measured by MST, which showed an increased affinity with a K_D of 152 nM. The different binding affinities (K_D) against different IRE1 α states could be explained by that IA107 prefers to bind to the dimeric form of IRE1 α to inhibit its RNase activity. The p-IRE1 α assembles more dimers and therefore has a higher affinity to IA107 in comparison with unphosphorylated IRE1 α . This may also explain the observed discrepancy between the inhibitory IC_{50} and the binding K_D of IA107. Once the dimeric proteins are bound and inhibited, the RNase activity of IRE1 α will be inhibited since the monomeric proteins are inactive.”

The allosteric inhibition mode induced by conformational changes upon binding may also be attributed to the discrepancy, IRE1 α may maintain the inactive conformation even after IA107 dissociation. The dissociated IA107 recycles and binds to other IRE1 α , inducing an inactive conformation and contributing to enhancing the inhibitory activity. The sentence in the previous manuscript (lines 333-335, page 17) kindly referred by the Reviewer has now been rewritten on page 20 of the revised manuscript as:

“IA107 may preferentially bind to the active dimeric IRE1 α , which could cause the different affinities against p-IRE1 α and IRE1 α and the observed discrepancy between the binding affinity and functional inhibitory potency. The discrepancy may also be attributed to the allosteric inhibition mechanism of IA107, which binds to the kinase domain and induces conformational changes to inhibit the RNase activity rather than directly competing with the RNA substrate. After the dissociation of IA107, IRE1 α may remain in the inactive conformation, contributing to the sustained inhibition effect.”

The aforementioned figures stand as:

Fig. 5e: IA01, IA06, IA10, IA64, and IA107 against the kinase activity of IRE1 α in the LanthaScreen™ Eu kinase binding assay (Thermo Fisher SelectScreen). **Fig. 4g:** IA107 binding affinity towards p-IRE1 α in MST assay.

3. One last point that require additional experiments is the cell validation. Currently, AI107 efficacy has only been validated in one cell line, A549, whereas its cytotoxicity was evaluated in A549 and three additional cancer cell lines (MDA-MB-468, HCT116 and HT-29). When making such bold statement as “Despite the reported examples, the current collection of IRE1 α inhibitors suffered from poor activity and unfavorable selectivity, which limited the further progression of such inhibitors” (L60-62, p4), the least would be to validate much more in depth your lead compound. Putting down other people’s work is never a good look, especially when there is a molecule in phase II clinical trial (ORIN1001/MKC8866) or other very selective and potent IRE1 kinase inhibitors (Braun et al. 2024, JMedChem) Without going as far as requiring ADME and in vivo validation, more testing on other cell lines needs to be done.

We thank the Reviewer’s comments on the cell validation and reported IRE1 α inhibitors and apologize for the inaccurate description here. We have now revised the description on page 4 as:

“Despite the reported IRE1 α inhibitors with impressive potency and encouraging progress, a novel inhibitor chemotype with a distinct inhibitory mechanism may offer a new therapeutic option in addressing related diseases.”

The activity of IAPD1 against the ER stress-induced *XBPI* mRNA splicing and the *IRE1 α* transcription level has now been further evaluated in HCT 116 and HT-29 cell lines, the result has been added to the revised manuscript as Extended Data Fig. 7e-h. Correspondingly, the following description has been added in the revised manuscript on page 18 as:

“The activity of IAPD1 against the ER stress-induced *XBPI* mRNA splicing and the *IRE1 α* transcription level were further evaluated in HCT 116 and HT-29 cell lines (Extended Data Fig. 7e-h). Consistent results were observed in all three tested cell lines. IAPD1 treatment inhibited the *XBPI* mRNA splicing with IC₅₀ of 220 nM and 160 nM in HCT 116 cells and HT-29 cells, respectively (Extended Data Fig.7e, 7g), with no obvious impact on the transcription level of *IRE1 α* (Extended Data Fig.7f, 7h).”

Additionally, the downstream XBP1s protein level upon IAPD1 treatment was evaluated by western blot in A549 cells and MDA-MB-231 cells, the result of which has been added to the revised manuscript as Fig 7d and Extended Data Fig.7i. Correspondingly, the following description has been added in the revised manuscript on page 18 as:

“The translated XBP1s protein and IRE1 α protein levels were evaluated via western blot in both A549 and MDA-MB-231 cell lines (Fig. 7d, Extended Data Fig.7i). KIRA8 and G1749 were used as reference compounds⁴². Under tunicamycin stimulation, the XBP1s protein concentration increased in both cell lines. The pre-treatment of IA107 and KIRA8 inhibited the ER stress-induced XBP1s protein increment. Meanwhile, treatment of G1749 at either 1 μ M or 5 μ M partially inhibited the XBP1s protein induced by ER stress. The treatment of compounds IAPD1, KIRA8, and G1749 did not affect the IRE1 α protein level, indicating that the observed activity of the compounds against IRE1 α downstream proteins was not due to the direct effect on the IRE1 α protein as degraders.”

The aforementioned figures stand as:

Extended Data Fig. 7e: IAPD1 dose-dependently inhibited ER stress-induced *XBP1* splicing in HCT 116 cells. Cells were pre-treated with IAPD1 for 2 h and a final concentration of 0.5 μg/mL tunicamycin was added into the cells and incubated for 2 h to induce the ER stress. **Extended Data Fig. 7f:** IAPD1 treatment shows minimal impact on the *IRE1α* transcription level in HCT116 cells. **Extended Data Fig. 7g:** IAPD1 dose-dependently inhibited ER stress-induced *XBP1* splicing in HT-29 cells. **Extended Data Fig. 7h:** IAPD1 treatment shows minimal impact on the *IRE1α* transcription level in HT-29 cells. **Fig.7d, Extended Data Fig. 7i:** IAPD1 inhibited the XBP1s protein level in A549 cells and MDA-MB-231 cells by Western blot assay. Cells were treated with indicated compounds for 2 h, then a final concentration of 1μg/mL tunicamycin was added into the cells and incubated for another 4 h before cells were lysed.

Minor points:

Most of the ITC experiments with the IA compounds do not capture the full S curve and could use some optimization of the conditions. Playing with the concentrations of the protein or drugs and doing dual injection runs might be worth pursuing. Additionally, please provide the thermodynamic data of the ITC runs.

We thank the Reviewer for the comments on the ITC experiments. We agree that adjusting the conditions could improve the S curve for some of the compounds. In the manuscript, we used

the same condition to have a better comparison across the compounds. Some compounds do not have the full S-curve, partially due to the weaker binding affinities. We have titrated 200 μM compound IA107 to 20 μM dephosphorylated IRE1 α , which gave a smoother S curve with a fitting K_D of 1.41 μM (similar to the one we presented in the manuscript by titrating 200 μM IA107 to 10 μM IRE1 α , 0.94 μM) (Fig R6). To obtain the full S curve for every compound would need to use different concentrations based on their varied affinities, however, to have a consistent condition across all ITC experiments for ease of comparison, we adopted the condition to titrate 200 μM compounds into 10 μM proteins for all ITC runs.

The thermodynamic data for the ITC experiments has been added to the revised manuscript Fig. 2f, Fig. 4e, Fig. 5b, Extended Data Fig. 3i-k, and Extended Data Fig. 4b-d.

The aforementioned figures stand as:

Fig. R6 200 μM IA107 to 20 μM IRE1 α

Fig. 2f

Fig. 4e

Fig. 5b

Extended Data Fig. 3i**Extended Data Fig. 3j****Extended Data Fig. 3k****Extended Data Fig. 4c****Extended Data Fig. 4d**
Please find below some minor points to the authors. I've also provided an annotated manuscript with comments regarding minor issues to address such as typos, references to add, questions, etc. The scheme of IA84 in the SI is incorrect (it is KIRA8).

We thank the Reviewer for the minor points annotated in the commented manuscript. The kindly pointed out issues have now been corrected in the revised manuscript, including the correction of the IA84 structure scheme in the SI.

POINT-BY-POINT RESPONSE

Reviewer #2

The two major concerns for me were that the authors hadn't considered an alternative interpretation that the small shifts in the structure could be due to crystal packing. They have now added a statement to the manuscript and have tried to justify the structures by analysing all four molecules within the asymmetric unit.

The second concern was the in vitro analysis of IRE1 oligomerisation was conducted using protein that was devoid of the luminal domain which is a driver for oligomerisation. Although cross linking in cells was attempted, they were not greatly successful. The use of the cytosolic domain only has now been added as a discussion point with implications for mechanism.

Response:

We are grateful to the Reviewer for the comments on the revised parts of the submitted manuscript regarding the two major concerns that were raised previously.

Reviewer #3

I am pleased to see that the authors of the manuscript "Harnessing Indole Scaffolds to Identify Small-molecule IRE1 α Inhibitors Modulating XBP1 mRNA Splicing" carried out all the experiments requested by the reviewers, which shared many similar concerns. I don't have major comments anymore, only two minor points:

1- There is a mistake in the structure of Figure 1, panel b. In the bottom right corner of the kinase binder area, it is not temozolomide but Z4P, a ligand reported by Pelizzari-Raymundo in 2023 in iScience.

Response:

We would like to thank the Reviewer for the general comments acknowledging the experiments performed in the revision to address the Reviewers' comments.

For the first point kindly mentioned by the Reviewer regarding the mistake in the structure in Figure 1, panel b: We have now provided a revised version of panel b (as shown below) with the corrected compound name of "Z4P". We have also carefully reviewed all other structures throughout the manuscript regarding compound names and the accuracy of the structures.

2- The thermodynamic data in figure 2f of the titration of IA01 are incoherent (N , ΔH and $-T\Delta S$). Something is not right with that experiment.

Once these two points are addressed, I am in favor of publication.

Response:

We thank the Reviewer's comments on the thermodynamic data of IA01 (Fig. 2f). The N , ΔH , and $-T\Delta S$ values of IA01 were significantly different from those of the other tested compounds, which might be attributed to the early saturation of the titration that resulted in a very small N value and a very high ΔH value.

After manually fitting the data, fitting N to 1 (as we observed a 1:1 binding mode based on the crystal structure), we obtained a K_D (2.62 μM) that is equivalent to the IC_{50} (2.47 μM) obtained from the kinase displacement assay, together with a more coherent ΔH and $-T\Delta S$. Thus, we updated the data with the new fitting curve in the resubmitted version of Figure 2. Additionally, we added the following description regarding the analysis information of this compound in the Method section for ITC titration as: "The one set of sites fitting model is used for the calorimetry data analysis, with cell concentration and syringe concentration as the input parameters. For compound IA01, the N was set to 1 to improve the fit".